# Air temperature changes in SW Greenland in the second half of the 18th century

Rajmund Przybylak[1,2], Garima Singh[1], Przemysław Wyszyński[1,2], Andrzej Araźny[1,2], Konrad Chmist[1]

[1]Nicolaus Copernicus University, Faculty of Earth Sciences and Spatial Management, Lwowska 1, PL-87-100 Toruń, Poland
[2]Centre for Climate Change Research, Nicolaus Copernicus University in Toruń, Poland

*Correspondence to*: Przemysław Wyszyński (Przemyslaw.Wyszynski@umk.pl)

**Abstract.** The thermal conditions of south-western Greenland in the second half of the 18th century were estimated using two unique series of meteorological observations. The first series (Neu-Herrnhut, 1st Sep 1767 to 22nd Jul 1768, hereinafter 1767–68) is the oldest long-term series of instrumental measurements of air temperature available for Greenland. The second (Godthaab, Sep 1784 to Jun 1792) contains the most significant and reliable data for Greenland for the study period. The quality controlled and corrected data were used to calculate daily, monthly, seasonal and yearly means. The daily means were further used to calculate day-to-day temperature variability (DDTV), thermal seasons, growing degree days (GDD), air thawing index (ATI), positive-degree days (PDD) and air-freezing-index-degree days (AFI).

Air temperature in Godthaab (now Nuuk) was, on average, warmer than the present day (1991–2020) in 1767–68 and colder in 1784–92. Compared to the Early Twenty Century Arctic Warming (ETCAW) period, the data for the two sub-periods show that the late 18th century was as warm or even warmer. Except winter 1767/68, winters and springs in the study period were longer, while summers and autumns were shorter than at present. The analysed climate indices usually do not exceed the maximum and minimum values from 1991–2020. Mean daily air temperature in studied historical periods rarely exceed $\pm 2$ SD of the long-term mean calculated for the contemporary period. Their distribution was usually close to normal, both in historical and contemporary periods.

## 1 Introduction

As we know, the Arctic plays a critical and leading role in the climate change we are now observing. The changes in climate and environment are greatest here (IPCC 2021). To precisely estimate the scale of influence of human activity on the Arctic climate, the range of natural climate changes should first be estimated. According to existing knowledge, little anthropogenic influence was noted in the Arctic before the mid-20th century. For this period, regular instrumental meteorological observations are very sparse, short and incomplete, especially before the 1920s (Przybylak 2000, 2002a, 2016; van Wijngaarden 2015; Brönnimann et al. 2019). For a longer perspective, all available early-instrumental observations should be collected and used. These were made mainly during numerous exploratory and scientific expeditions to the Arctic. We have gathered and processed this kind of data in our previous works (Przybylak 2000; Przybylak and Vizi 2005; Przybylak et al.

2010, 2016, 2021, 2022; Nordli et al. 2014, 2020; Przybylak and Wyszyński 2017). In those works, we focused on climate analysis for a time span including the 19th century and the first half of the 20th century.

The cited papers, however, did not analyse meteorological observations made in Greenland by the Moravian missionaries beginning in the second half of the 18th century. According to our knowledge, only in a small number of recently published papers (Vinther et al. 2006; Demarée et al. 2020; Demarée and Ogilvie 2021) have limited analyses of some series

of instrumental observations covering the 18th century been performed for the study area. In the first paper, monthly means from Nuuk from the period 1784–92 were used. According to the authors, these monthly means organised into tables were archived at the Climate Research Unit (CRU) at University of East Anglia by Hubert Lamb, who visited the Danish Meteorological Institute (DMI) in the mid-1970s and made photocopies of them (P. Jones, pers. comm.). Data were gathered and elaborated by the staff of the DMI under the leadership of Knut Frydendahl. Summer, winter and annual mean values are

shown in Fig. 5 of the Vinther et al. paper and were used, together with spring and autumn means (not shown in the paper), for construction of the merged SW Greenland instrumental temperature series (see their Figure 10). On the other hand, Demarée et al. (2020), and Demarée and Ogilvie (2021) briefly analyse data available for Neu-Herrnhut (now Nuuk) for the period September 1767 to July 1768. Figures presented in both publications show originally published sub-daily air temperature (8 a.m. and 2 p.m.) simply converted from Fahrenheit scale to Celsius scale and daily pressure values converted

from Paris inches and lines into the presently used unit, i.e. hectopascals (hPa).

Additionally, in recent years, the possibility of using old individual measurements published in annual reports (diaries) made by the Moravian missionaries (handwritten in old German) for climate studies, but particularly for the study of the weather extremes in Greenland, was presented by Kodzik (2019) and Born et al. (2021). Both publications, as well as Lüdecke (2004), Demarée and Ogilvie (2008) and Demarée et al. (2020), present detailed histories of meteorological

measurements made by Moravian missionaries in Greenland. For this reason, we omitted this information here.

At the end of this short review of the state of the art of the knowledge about weather and climate in Greenland in the second half of the 18th century in light of the available instrumental observations, we want also to indicate some of the most important historical print publications analysing very briefly and fragmentarily some aspects of weather in Greenland and attaching some instrumental meteorological data. Firstly, we should mention a very well-known publication written by the

Moravian missionary David Cranz (1723–77) entitled History of Greenland (German ed. 1765, Engl. ed., spelling the author's name Crantz, 1767) and its supplement (Cranz 1770). In the latter publication, Cranz published some limited set of data from Neu-Herrnhut selected from the expedition year 1767/68 sent to him by Christopher Brasen (1738–74). In the same year as Cranz, i.e. 1770, so too Kratzenstein (1723–95), professor of experimental physics and medicine at the University of Copenhagen, published data from Brasen's meteorological observations. In 1820, the updated version of Cranz' history of

Greenland was published in which selected meteorological data from the period 1767–68 are also available. Finally, we also need to mention the publication of Danish Reverend Andreas Ginges (1754–1812), who made meteorological observations in Godthaab from 1782 to 1792. Only a short series of his observations (October 1786 to June 1787) was published in the 1787 Ephemerides Societatis Meteorologicae Palatinae Society yearbook (Ginge 1789).

In this paper we present a comprehensive analysis of thermal conditions in SW Greenland in the second half of the 18th century based on all available sub-daily measurements made by the Moravian missionaries. The early-instrumental meteorological data that we analyse are the oldest series that exist not only for the Greenland but also for the entire Arctic. The main aim of the paper is to reconstruct thermal conditions in Greenland in the second half of the 18th century using different measures and to estimate differences in comparison to the present climate in the region. Such knowledge is crucial for the validation of temperature reconstructions based both on modelling works and on various proxies, because these show inconsistent results. For example, Kobashi et al. (2010) estimate that the temperature in central Greenland in the second half of the 18th century was the coldest in the entire millennium. It was about 2 °C colder in comparison to both present data and the data for the Medieval Warm Period (MWP), particularly for the 12th century (see their Fig. 13). On the other hand, a recently published reconstruction of central and northern Greenland temperatures stacked from a compilation of 21 stable oxygen isotope records ($\delta^{18}$O anomalies relative to the 1961–90 reference interval) revealed the existence of a warm period in the second half of the 18th century of comparable magnitude to that occurring during the MWP, but colder than the recently observed temperature (see Fig. 1a in Hörhold et al. 2023).

## 2 Area, data and methods

### 2.1 Area and data

The only meteorological data available for Greenland in the 18th century come from its SW part, and more specifically from the area where the present capital of Greenland, Nuuk (older used names: Godthaab, Godthåb or Godthab) is located (Fig. 1). A mission and trading station were established in this place by Hans Egede in the 1720s (Mills 2003). Nuuk is presently both the capital and main port of Greenland. It is located on a peninsula (~70 km long) on the south-western coast at the mouth of Godthab Fjord close to mountains such as Sermitsiaq and Hjortetakken (Nuttal 2005). According to the Köppen–Geiger climate classification (Kottek et al. 2006, see also http://koeppen-geiger.vu-wien.ac.at/), the climate in Nuuk is a tundra climate type (ET) and part of the polar climate zone. The average annual air temperature here in 1991–2020 was -1.0 °C. The mean long-term temperature of the warmest month (July) was positive (7.0 °C) but did not exceed 10.0 °C in individual years. The coldest month is February with an average monthly air temperature of -8.3 °C. The average annual total of precipitation at Nuuk in period 1991–2020 was 874.0 mm. Sums of precipitation were highest in September (106.0 mm) and lowest in April (53.0 mm) (Cappelen and Drost Jensen 2021; for more details, see https://www.dmi.dk/fileadmin/Rapporter/2021/DMI_report_21_12_Greenland.pdf).

As results from Fig. 1, the historical site (No. 1) is located very close to the present ones (Nos 2 and 3) and these are all coastal stations. For these reasons, there is no need to introduce the kinds of corrections that would be required if the sites were in different locations from one another.

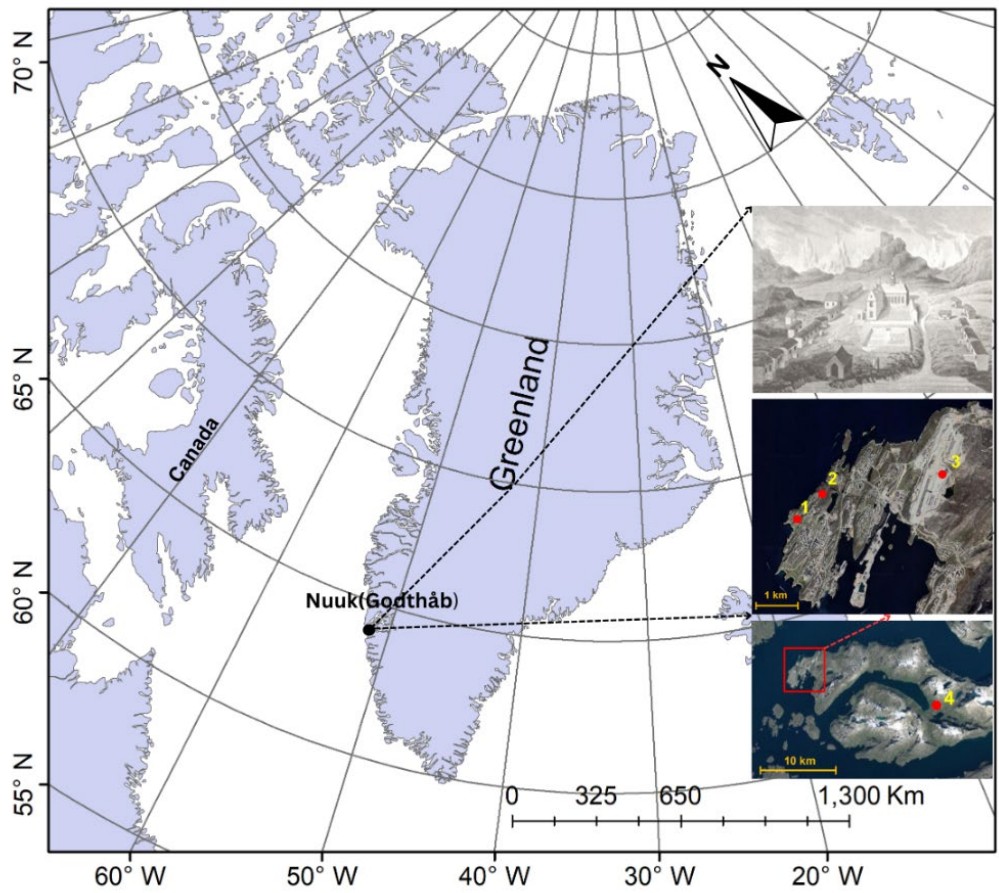

**Figure 1:** Study area and location of historical and contemporary sites of meteorological measurements. Explanation: 1 – historical sites: Neu-Herrnhut (1767–68), Godthaab (1784–92); 2 – 4250 Nuuk (1991–2020); 3 – 4254 Mittarfik Nuuk (2001–20); 4 – Pissiksarbik. Upper photo: source Neu-Herrnhut (Longman, Hurst, Rees, Orme & Brown, 1818). Map data for location of sites: Google Earth; Images © 2023 Maxar Technologies, © 2023 Airbus and © 2023 Asiaq.

To study the thermal conditions of SW Greenland in the second half of the 18th century, the two available series of meteorological observations have been used: (1) for Neu-Herrnhut (1 Sep 1767 to 22 Jul 1768, gap from 4 to 24 October, 1767) and (2) for Godthaab (Sep 1784 to Jun 1792, with some gaps; for details, see Fig. 2). This earlier period for which data is available is referred to in this article as "1767–68", distinguishing it from references to the expedition year, winter, or threshold spanning the two calendar years, which are described using a slash as "1767/68". In analysing this series of data it is important to note that the place of observations was changed in the beginning of June 1768 to Pissiksarbik (Cranz 1820), which is located about ten miles to the east from Neu Herrnhut (see Fig. 1). As results from Fig. 2, in the period 1784–92,

110 there are a lot of gaps. However, for three expeditions years we have at least ten months of data. The longest continuous series is available for period Jan 1790 to Jun 1792. The reasons of the existing gaps in this period are not known. Examples of manuscripts presenting meteorological observations for these two series are shown in Fig. 3.

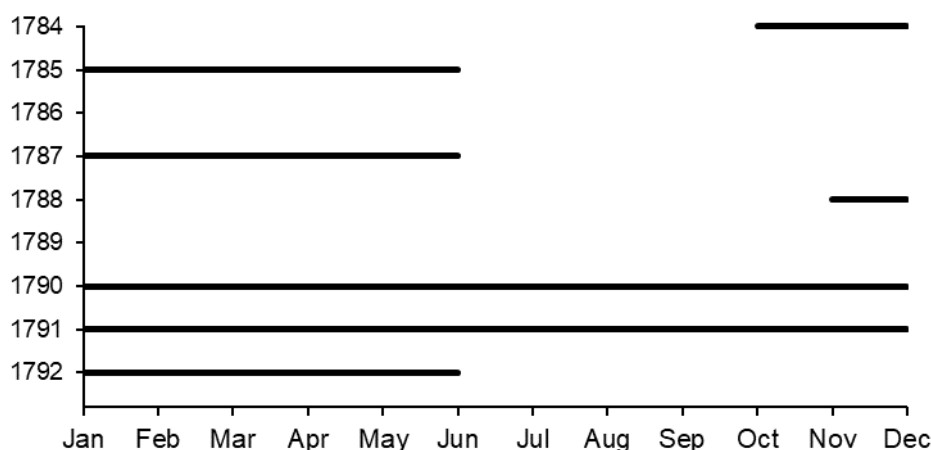

115 **Figure. 2**: Coverage of air temperature data in Nuuk (orig. Godthaab), 1784–92.

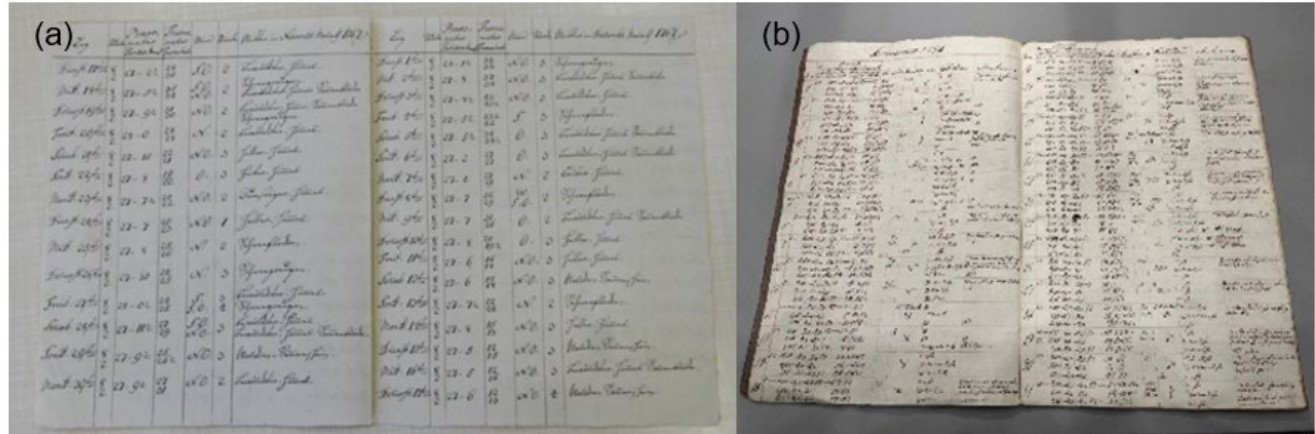

**Figure 3:** Examples of manuscripts presenting meteorological observations: (a) for Neu-Herrnhut (1 Sep 1767 to 22 Jul 1768), source: MH R.15 J.a.13.9. Data presented in the manuscript: 17 November to 17 December 1767, (b) for Godthaab (1782–92), source: Astronomiske og
120 meteorologisk Iagttagelser, anstillede i Godthaab i Grønland 1782–92 (Det Kgl. Bibliotek in Copenhagen). Data presented in the manuscript: January 1791.

The first series, as we already mentioned, is the oldest long-term series of instrumental measurements of air temperature. In addition, the weather register (Moravian Archive in Herrnhut, catalogue number R.15.J.a.13.) provides more

measurements such as wind direction (from 8 directions) and force (on a scale from 1 to 6), as well as a very short weather description. Meteorological observations were made by Christopher Brasen (1738−74) usually two times a day – at 8 a.m. and 2 p.m. According to Borm et al. (2020), Brasen received a thermometer (and a barometer) from Christian Gottlieb Kratzenstein (1723–95), Rector of the University of Copenhagen. The second series of measurements, although not continuous (see Fig. 2), is the greatest and most reliable available for Greenland for the study period. Observations were made three times a day (7 a.m., 2 p.m. and 9 p.m.) by the Danish Reverend Andreas Ginges (1754–1812) using a methodology and instruments provided by the Meteorological Society of the Palatinate. The sub-daily or daily air temperature data exist for the following periods: Sep 1784 to Jun 1785, Jan−Jun 1787, Nov−Dec 1788, Jan 1790 to Jun 1792 and are available in the manuscript entitled "Astronomiske og meteorologisk Iagttagelser, anstillede i Godthaab i Grønland 1782–1792" and in the society's yearbook Ephemerides Societatis Meteorologicae Palatinae, which contains data only for 1787. Unfortunately, information about precise place of measurements, sheltering and exposure, producers of thermometers, etc. is not available for either series.

## 2.2 Methods

Climatological studies based on historically recorded data are burdened with the hard reality that the data we uncover is inevitably imperfect. This fact is counterbalanced by the high value of the data that stems from its rarity and great potential to inform us about the climate of the past. Statistical techniques are thus used both to compensate for these quality issues and to take into account the uncertainty that these compensations introduce.

Therefore, all available historical data (Singh et al. 2023, see link to database https://doi.org/10.18150/L1Y21Q) were quality controlled and corrected prior to being used to statistical analysis. In the first step, in order to avoid errors, data were extracted manually into digital format. This kind of approach guarantees the lowest error rates when acquiring data from antique books, prints and unpublished sources (Brönnimann et al. 2006). In the second step, all outliers were checked to estimate the probability of their occurrence. Suspect values were not found. In the third step, the original measured values were converted to the presently used unit, i.e. degrees Celsius (for example, air temperature in the period 1767–68, which was measured using a Fahrenheit thermometer). In the fourth step, mean daily air temperature (MDAT) values were calculated using available sub-daily temperature. For the expedition year 1767/68, measurements were made mainly at 8:00 and 14:00, but there were also days when three measurements a day were taken (the third at 22:00 or 23:00). On the other hand, stable hours of measurements (7:00, 14:00 and 21:00) were used in the period 1784–92. As a result, we calculated MDAT for the historical periods using the following formulas:

$$MDAT = (T8+T14)/2 \tag{1}$$

$$MDAT = (T8+T14+2*T22)/4 \tag{2}$$

$$MDAT = (T8+T14+2*T23)/4 \tag{3}$$

$$MDAT = (T7+T14+2*T21)/4 \tag{4}$$

For the contemporary period (1991–2020) the hourly data are available for 4250 Nuuk station (Drost Jensen (ed.)
2022, data available at https://www.dmi.dk/publikationer/), and therefore the MDAT was calculated using the following formula:

MDAT= (T1+T2+T3+ … T24)/24 (5)

Then, data for the few gaps in MDAT (6.4%) for 4250 Nuuk station were interpolated from the neighbouring 4254 Mittarfik Nuuk station by the least square method, which is explained in detail by Nordli et al. (2020). This raised the final
completeness of the series of MDAT for 4250 Nuuk station for period 1991–2020 to 99.7%.

Furthermore, the scale of the influence that the different measurement times had on the values of MDAT was checked using hourly air temperature data taken from 4250 Nuuk station for the contemporary period (2001–10). Corrections needed to obtain "real" MDAT (calculated according to formula 5) were calculated for each month separately (see Table 1). It is clear that the time of temperature observations has a greater influence on the MDAT in the warm half-year than in the cold half-
year. Thus, all historical MDAT values were corrected using the values shown in Table 1.

**Table 1:** Air temperature corrections (°C) used to calculate MDAT from the historical periods

| Formula | J | F | M | A | M | J | J | A | S | O | N | D |
|---|---|---|---|---|---|---|---|---|---|---|---|---|
| MDAT 1-MDAT5 | -0.004 | 0.135 | 0.078 | 0.205 | 0.334 | 0.507 | 0.525 | 0.461 | 0.263 | 0.068 | 0.033 | -0.005 |
| MDAT 2-MDAT5 | -0.017 | 0.052 | -0.007 | 0.031 | -0.014 | 0.003 | -0.053 | -0.020 | -0.010 | -0.007 | -0.042 | -0.021 |
| MDAT 3-MDAT5 | -0.002 | 0.031 | -0.003 | 0.002 | -0.080 | -0.135 | -0.249 | -0.137 | -0.059 | -0.043 | -0.022 | -0.034 |
| MDAT 4-MDAT5 | 0.000 | 0.031 | -0.040 | 0.034 | 0.090 | 0.122 | 0.115 | 0.068 | -0.001 | -0.018 | -0.020 | -0.018 |

Thus corrected, MDATs do not contain biases connected with differences in measurement times. The two other possible
potential types of biases connected with (1) exposition of thermometers and (2) their accuracy could not be corrected by us because of the lack of information. We should add here, however, that the first bias (associated with wrong exposition of thermometers) may have influenced the temperature measurements, but mainly during the polar day.

The corrected MDAT values (which are available at https://doi.org/10.18150/L1Y21Q, Singh et al. 2023) were used to calculate standard (monthly, seasonal and annual means) and less typical climate statistics (indices) such as day-to-day
temperature variability (DDTV), thermal seasons, growing degree days (GDD), air thawing index (ATI), positive-degree days (PDD) and air-freezing-index-degree days (AFI). The last four indices were calculated using definitions proposed by Nordli et al. (2020) (see also Table 2). They are less known and used in climate analysis, so we briefly summarise their importance in studies of climate and of environment. For example, the GDD index (or number of growing days) significantly "impacts plants' and animals' activity and growth which in the Arctic region may start as soon as snow melting has taken place" (Nordli
et al. 2020). On the other hand, the ATI index is often used in permafrost engineering, engineering design and for estimations of active layer thickness above the permafrost (Instanes 2016). The PDD index, in turn, is also commonly used by glaciologists, such as for modelling glacier or snow melt, which is possible only when temperature is above 0 °C. The PDD can therefore be

thought of, according to Huybrechts and Oerlemans (1990), as the total energy available for melting snow and ice over the course of one year. The temperature oscillation around the 0 °C threshold is also extremely important for studying, for example,

mechanical and chemical weathering processes in the Arctic. Also of importance is the fact that we calculated all of these indices and presented the results in the paper describing temperature change in Svalbard in the period 1898–2018 (see Nordli et al. 2020). This means that it is possible to compare the results between the pre-anthropogenic period (before *ca* 1950 for the Arctic) and recent warming in both areas (i.e., Greenland and Svalbard).

**Table 2:** Definitions of terms used in threshold statistics (after Nordli et al. 2020)

| Terms | Definitions |
|---|---|
| Annual growing degree-days sum | $GDD = \sum_{i=1}^{n} Max(0, Ti - 5) \, for \, May - Sep$ |
| Air thawing index degree-days sum | $ATI = \sum_{i=1}^{n} Max(0, Ti) \, for \, May - Sep$ |
| Positive degree-days sum | $PDD = \sum_{i=1}^{n} Max(0, Ti) \, for \, Oct - Apr$ |
| Air freezing index degree-days sum | $AFI = \sum_{i=1}^{n} Min(0, Ti) \, for \, Oct - Apr$ |
| *Ti* | Mean temperature on day *i* and *n* is the number of days |
| *n* | Number of days |

To describe the day-to-day MDAT variability (DDTV) we calculated modulus of MDAT change from one day to the next, and the results were also smoothed using the Gaussian filter.

Thermal seasons for Greenland were analysed according to the proposition given by Baranowski (1968).

The thermal seasons fulfil the following criteria:

1 winter: MDAT ≤ -2.5 °C

2 spring and autumn: -2.5 °C < MDAT < 2.5 °C

3 summer: MDAT ≥ 2.5 °C.

To determine these thresholds, the method proposed by Kosiba (1958) was applied. This method allows seasons to be

205    distinguished in 1-year series of MDAT. The first day of a given season was determined, after Kosiba (1958), as the day from which, onwards, more days fulfil the criteria of the new season than of the previous season.

To estimate if the air temperature distribution (shown as frequency of occurrence of MDAT in 1-degree intervals) is normal or not in the historical and contemporary periods, the skewness ($\gamma 1$) and kurtosis ($\gamma 2$) of analysed sets of air temperature data were calculated according to formulas recommended by von Storch and Zwiers (1999).

# 3 Results

## 3.1 Monthly resolution

Annual courses of historical (1767–68, 1784–92) air temperature have been shown on the background of 30-year means from the contemporary period 1991–2020 (Table 3, Fig. 4). It is clearly shown that average air temperature from Sep 1767 to Jun 1768 was warmer than today by as much as 1.5 °C.

The second studied historical period (1784–92), which is more representative that the first one was on average by 1.4 °C colder than today (in the period Sep–Jun), but winter was particularly cold (anomaly -2.9 °C) (see Table 3). It is also important to note that summers in the period 1784–92 (only two available) were slightly warmer than at present (anomaly 0.3 °C). Another important finding is the fact that almost all monthly mean temperatures lie within 1 standard deviation (SD) of the present means. Only in June and July 1768 did values of monthly means exceed 1 SD, but they were within 2 SD (Table 3, Fig. 4).

**Table 3:** Mean monthly, seasonal and annual air temperature and variability (SD, DDTV) of MDAT in Nuuk in the historical periods

| Mean temperature (°C) | | | | | | | | | | | | | | | | | |
|---|---|---|---|---|---|---|---|---|---|---|---|---|---|---|---|---|---|
| Period | S | O | N | D | J | F | M | A | M | J | J | A | SON | DJF | MAM | JJA | SEP-JUN |
| 1767-68 | 3.6 | | -4.2 | -2.6 | -3.6 | -8.7 | -3.9 | -0.9 | 2.1 | 8.4 | 9.7* | | | -5.0 | -0.9 | | -1.1 |
| 1784-85 | 0.8 | -3.2 | -8.3 | -3.1 | -12.3 | -12.5 | -3.7 | -5.5 | 0.1 | 3.5 | | | -3.6 | -9.3 | -3.0 | | -4.4 |
| 1786-87 | | | | | -11.2 | -10.8 | -9.0 | -0.6 | 2.1 | 8.3 | | | | | -2.5 | | |
| 1789-90 | | | | | -10.3 | -10.5 | -9.9 | -3.2 | 0.4 | 4.6 | 7.6 | 8.4 | | | -4.2 | 6.9 | |
| 1790-91 | 2.6 | -0.9 | -4.0 | -10.6 | -18.1 | -10.8 | -11.4 | -0.3 | -2.0 | 5.1 | 6.8 | 6.1 | -0.8 | -13.2 | -4.6 | 6.0 | -5.0 |
| 1791-92 | 5.7 | -0.4 | -4.6 | -7.9 | -9.7 | -8.5 | -9.1 | -2.4 | 0.5 | 3.6 | | | 0.2 | -8.7 | -3.6 | | -3.3 |
| 1784-92 | 3.1 | -1.5 | -5.6 | -7.2 | -12.3 | -10.6 | -8.6 | -2.4 | 0.2 | 5.0 | 7.2 | 7.2 | -1.4 | -10.0 | -3.6 | 6.5 | -4.0 |
| 1991-2020 | 3.9 | 0.2 | -3.2 | -5.4 | -7.4 | -8.5 | -7.7 | -3.2 | 0.9 | 4.7 | 7.2 | 6.8 | 0.3 | -7.1 | -3.3 | 6.2 | -2.6 |
| 1784-92 - 1991-2020 (diff) | -0.8 | -1.7 | -2.4 | -1.8 | -4.9 | -2.1 | -0.9 | 0.8 | -0.7 | 0.3 | 0.0 | 0.4 | -1.7 | -2.9 | -0.3 | 0.3 | -1.4 |
| SD (°C) | | | | | | | | | | | | | | | | | |
| Period | S | O | N | D | J | F | M | A | M | J | J | A | SON | DJF | MAM | JJA | SEP-JUN |
| 1767-68 | 1.7 | | 2.7 | 6.2 | 5.9 | 4.2 | 4.3 | 2.4 | 3.4 | 2.8 | 3.0* | | 2.3** | 5.5 | 3.4 | | 3.6 |
| 1784-85 | 2.6 | 3.9 | 5.5 | 3.7 | 6.0 | 8.6 | 6.7 | 4.0 | 2.4 | 1.1 | | | 4.0 | 6.1 | 4.4 | | 4.5 |
| 1786-87 | | | | | 3.7 | 2.5 | 5.4 | 3.0 | 2.8 | 2.0 | | | | | 3.7 | | |
| 1789-90 | | | | | 5.5 | 6.6 | 5.6 | 5.0 | 4.0 | 2.7 | 1.5 | 2.3 | | | 4.9 | 2.2 | |
| 1790-91 | 2.7 | 3.4 | 4.4 | 4.3 | 4.3 | 6.7 | 5.8 | 2.2 | 3.7 | 2.6 | 1.9 | 1.2 | 3.5 | 5.1 | 3.9 | 1.9 | 4.0 |
| 1791-92 | 2.1 | 3.0 | 4.1 | 5.2 | 5.2 | 5.1 | 4.2 | 4.1 | 3.2 | 3.3 | | | 3.1 | 5.2 | 3.8 | | 4.0 |
| 1784-92 | 2.3 | 3.2 | 4.2 | 4.9 | 5.1 | 5.6 | 5.3 | 3.5 | 3.3 | 2.4 | 2.1 | 1.8 | 3.2 | 5.5 | 4.0 | 2.0 | 4.0 |
| 1991-2020 | 1.9 | 2.7 | 3.7 | 4.8 | 5.3 | 7.2 | 5.8 | 4.3 | 3.6 | 2.5 | 2.0 | 2.1 | 2.8 | 5.8 | 4.6 | 2.2 | 4.2 |
| 1784-92 - 1991-2020 (diff) | 0.4 | 0.5 | 0.5 | 0.0 | -0.1 | -1.6 | -0.4 | -0.8 | -0.3 | -0.1 | 0.1 | -0.3 | 0.5 | -0.6 | -0.5 | -0.1 | -0.2 |
| DDTV (°C) | | | | | | | | | | | | | | | | | |
| Period | S | O | N | D | J | F | M | A | M | J | J | A | SON | DJF | MAM | JJA | SEP-JUN |
| 1767-68 | 0.8 | | 1.3 | 2.6 | 3.0 | 2.3 | 2.6 | 1.8 | 1.5 | 1.9 | 1.7* | | 1.1** | 2.6 | 2.0 | | 2.0 |

| | | | | | | | | | | | | | | | | | |
|---|---|---|---|---|---|---|---|---|---|---|---|---|---|---|---|---|---|
| 1784-85 | 1.2 | 2.1 | 3.0 | 2.6 | 3.6 | 4.8 | 3.1 | 2.4 | 1.2 | 1.4 | | | 2.1 | 3.7 | 2.2 | | 2.5 |
| 1786-87 | | | | | 2.4 | 1.7 | 2.7 | 2.3 | 1.3 | 2.2 | | | | | 2.1 | | 2.1 |
| 1789-90 | | | | | 3.4 | 3.1 | 4.2 | 2.2 | 1.1 | 2.1 | 1.4 | 1.1 | | | 2.5 | 1.5 | 2.7 |
| 1790-91 | 1.2 | 2.2 | 3.4 | 2.5 | 2.6 | 4.6 | 2.3 | 1.4 | 1.9 | 1.4 | 1.5 | 1.0 | 2.2 | 3.2 | 1.9 | 1.3 | 2.3 |
| 1791-92 | 1.6 | 1.7 | 2.3 | 3.9 | 3.0 | 2.8 | 2.5 | 3.1 | 2.1 | 1.2 | | | 1.9 | 3.2 | 2.6 | | 2.4 |
| 1784-92 | 1.3 | 2.0 | 2.9 | 3.0 | 3.0 | 3.4 | 3.0 | 2.3 | 1.5 | 1.7 | 1.6 | 1.1 | 1.8 | 3.2 | 2.2 | 1.4 | 2.3 |
| 1991-2020 | 1.1 | 1.4 | 1.8 | 2.1 | 2.3 | 2.4 | 2.4 | 1.8 | 1.4 | 1.6 | 1.6 | 1.3 | 1.5 | 2.3 | 1.9 | 1.5 | 1.8 |
| 1784-92 - 1991-2020 (diff) | 0.3 | 0.6 | 1.0 | 0.9 | 0.7 | 1.0 | 0.6 | 0.5 | 0.1 | 0.1 | -0.1 | -0.2 | 0.6 | 0.9 | 0.4 | 0.1 | 0.6 |

Key: * until 22nd July; ** excluding 4–24 October

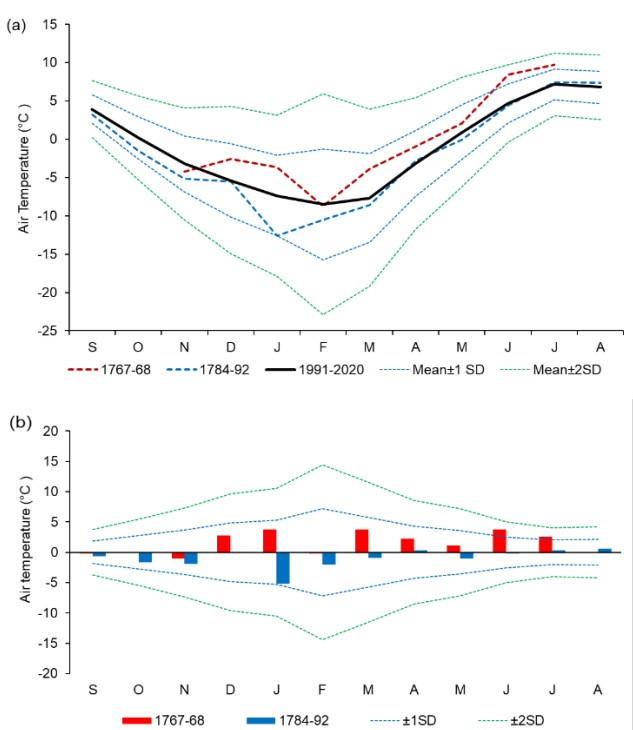


**Figure 4:** Annual courses of historical (1767–68, 1784–92) and modern air temperatures in Nuuk based on monthly means (a) and differences between them (b). SD has been calculated on the basis of present data (1991–2020)

265        The monthly averages of the DDTV in the historical periods were usually greater than at present – slightly greater in 1767–68 and much greater in all years from the period 1784–92 (Table 3). The differences were particularly large in the period from November to February (0.7–1.0 °C) and small (±0.2 °C) from May to September.

        The availability of MDAT allows us also to present the so-called threshold statistics for historical periods in Nuuk; these include GDD, ATI, PDD and AFI (Figs 5 and 6). The expedition year 1767/68 was very warm, and therefore GDD and
ATI for each month were equal to or higher than the average values of these indices observed in 1991–2020 (Fig. 5a, b). On the other hand, no important changes were observed for PDD, while AFI was usually lower than the present-day norm (Fig. 5c, d). The GDD and ATI in the period 1784–92 usually (except 1787) do not exceed the maximum and minimum values from 1991–2020 (Fig. 6a, b). The intensity of warm events (PDD) during the cold season (from October to April) in the period 1784–92 is close to the average and minimum PDD in 1991–2020 (Fig. 6c), but the AFI values in 1784–92 are between the
average and maximum AFI values calculated for 1991–2020 (Fig. 6d). The AFI is a measure of the magnitude and duration of sub-zero temperature events during the winter season each year.

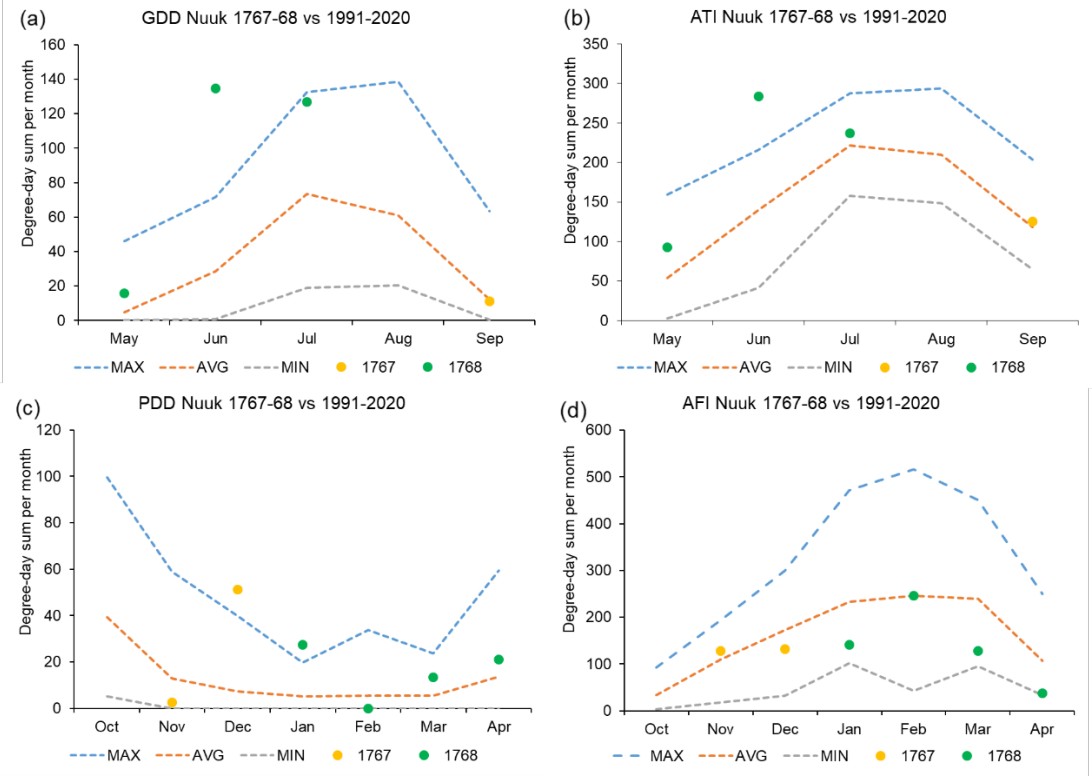

**Figure 5:** Comparison of temperature indices (GDD, ATI, PDD and AFI) calculated for Nuuk for historical (1767–68) and contemporary
(1991–2020) periods

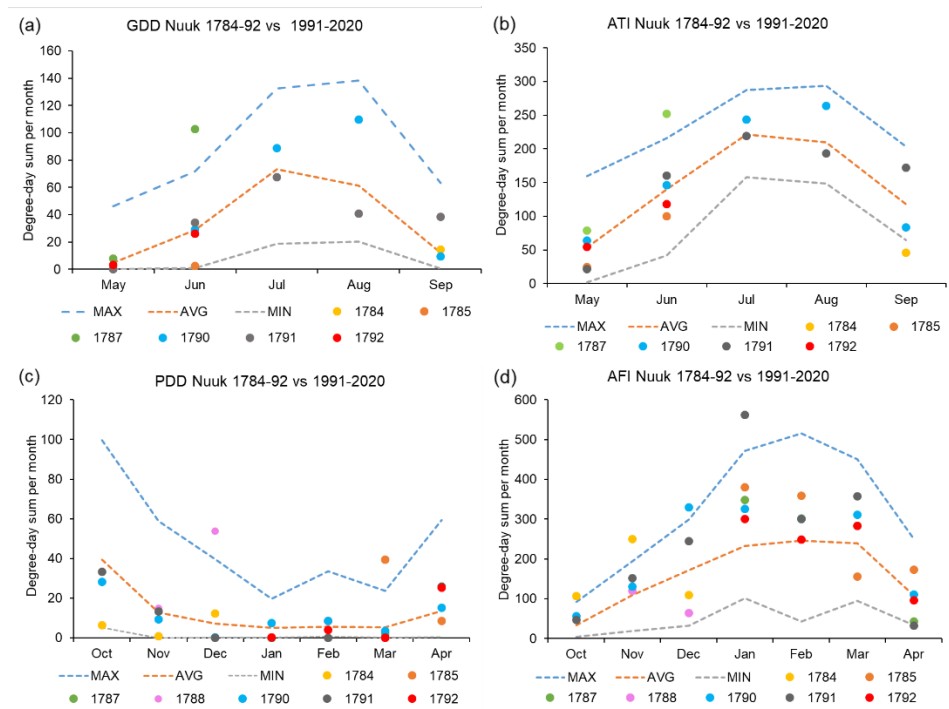

**Figure 6:** Comparison of temperature indices (GDD, ATI, PDD and AFI) calculated for Nuuk for historical (1784–92) and contemporary (1991–2020) periods

## 3.2 Daily resolution

In our earlier paper (Przybylak and Vízi 2005) analysing historical daily and sub-daily data for the Canadian Arctic we stated that: "In the process of averaging, important climatic information may very often be lost." That is why, as in the present paper, we decided to analyse the air temperature regime using data with greater-than-monthly resolution.

Annual courses in the historical period, based on MDAT available for SW Greenland, when superimposed on their present-day (1991–2020) mean annual courses, show that the study period included spells both warmer and colder than today (Fig. 7). As Fig. 7 shows for the historical period, we have six years for which a complete or near-complete annual cycle is available. Of all the years shown, 1767/68 was the warmest and 1790/91 the coldest (Table 3). In line with expectations, colder MDATs were markedly more common than warmer MDATs, especially in winter.

A particularly cold winter (DJF) occurred in 1790/91 when MDAT was 10–15 °C colder than the mean MDAT for the period 1991–2020 (Fig. 7). On the other hand, a long span of exceptionally large positive MDAT anomalies in relation to present-day values (up to about 10 °C) was observed at the turn of December to January 1767/68 (Fig. 7). So too, the summer in this year was significantly warmer than today. Average MDATs in summer in the period 1784–92 were usually slightly warmer than or, rarely, near the present norm, whereas in spring they were closer to present-day values, in particular in April

and May (Fig. 7). The greatest negative MDATs anomalies (exceeding 5 °C) were particularly noted in November, January and February (Fig. 7). As with mean monthly data, the majority of average MDATs from the period 1784–92 lie within 1 SD of the present means. On the other hand, most MDAT data in individual years do not exceed 2 SD from present-day means (Fig. 7).

More precise information about the character of air temperature changes between historical and contemporary periods is presented in Fig. 8, which shows relative frequencies of occurrence of air temperature stratified into one-degree intervals. As results from Fig. 8, MDATs in the four analysed seasons in Nuuk usually have a distribution close to normal in historical and contemporary periods alike (values of skewness [$\gamma1$] usually range between -0.5 and 0.5). The distribution decidedly most close to normal is noted for summer 1784–92. These summers also exhibited the smallest changes in MDAT distributions between historical and present periods (Fig. 8b). On the other hand, the largest changes in MDAT distributions clearly occurred in summer 1767–68 (Fig. 8a, there are more cold intervals at present) and autumn and winter 1784–92 (Fig. 8b, there are more warm intervals at present). It is also very clearly shown that the MDAT data are not heavily-tailed; small negative values of kurtosis prevail ($\gamma2$, i.e., platykurtic distribution), especially in the period 1784–92.

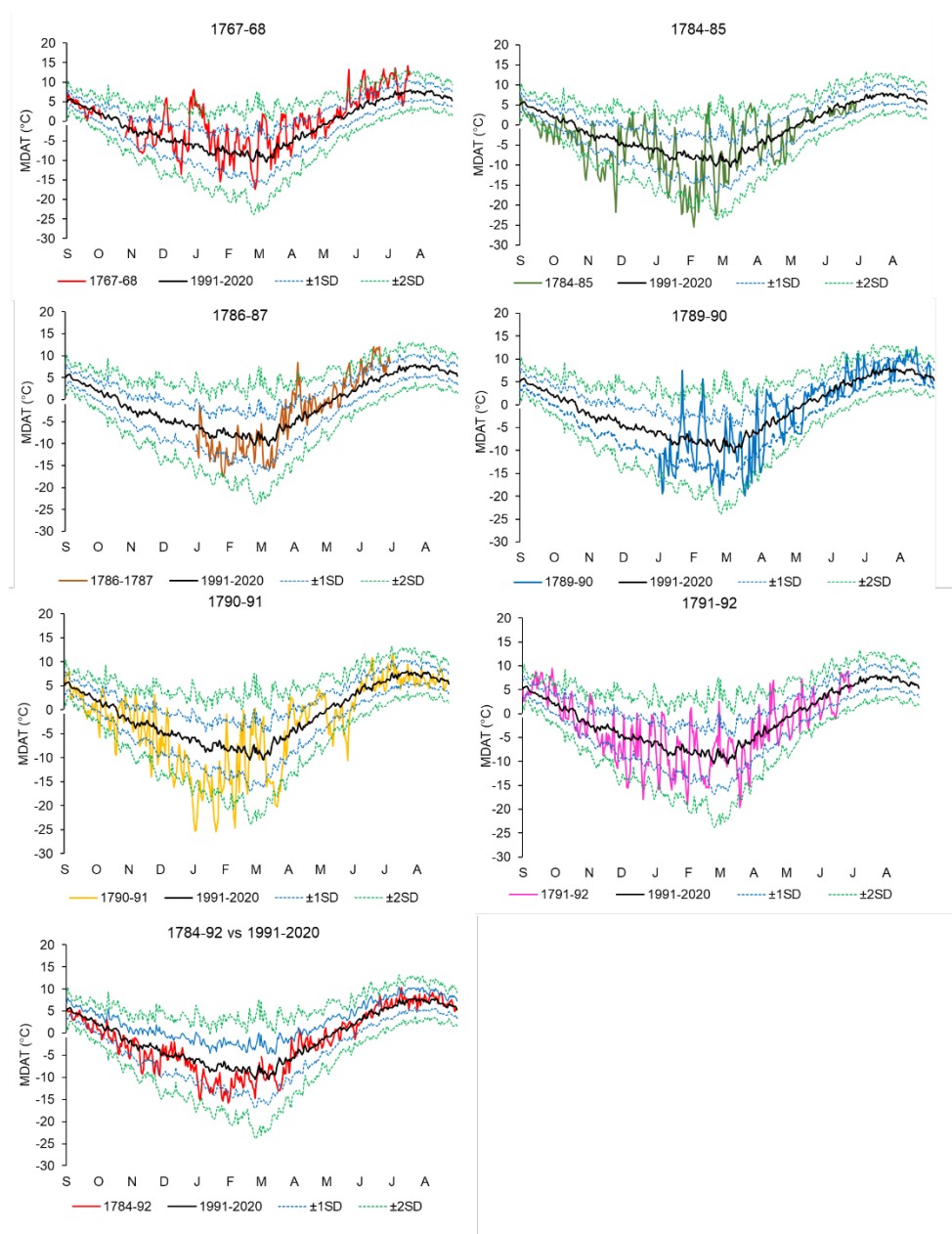

**Figure 7:** Annual courses of MDAT in Nuuk in historical years (lines in different colours) and 1991–2020 mean (black line). Blue (±1 SD) and green (±2 SD) dashed lines indicate SD calculated for 1991–2020 and added/subtracted from the present mean

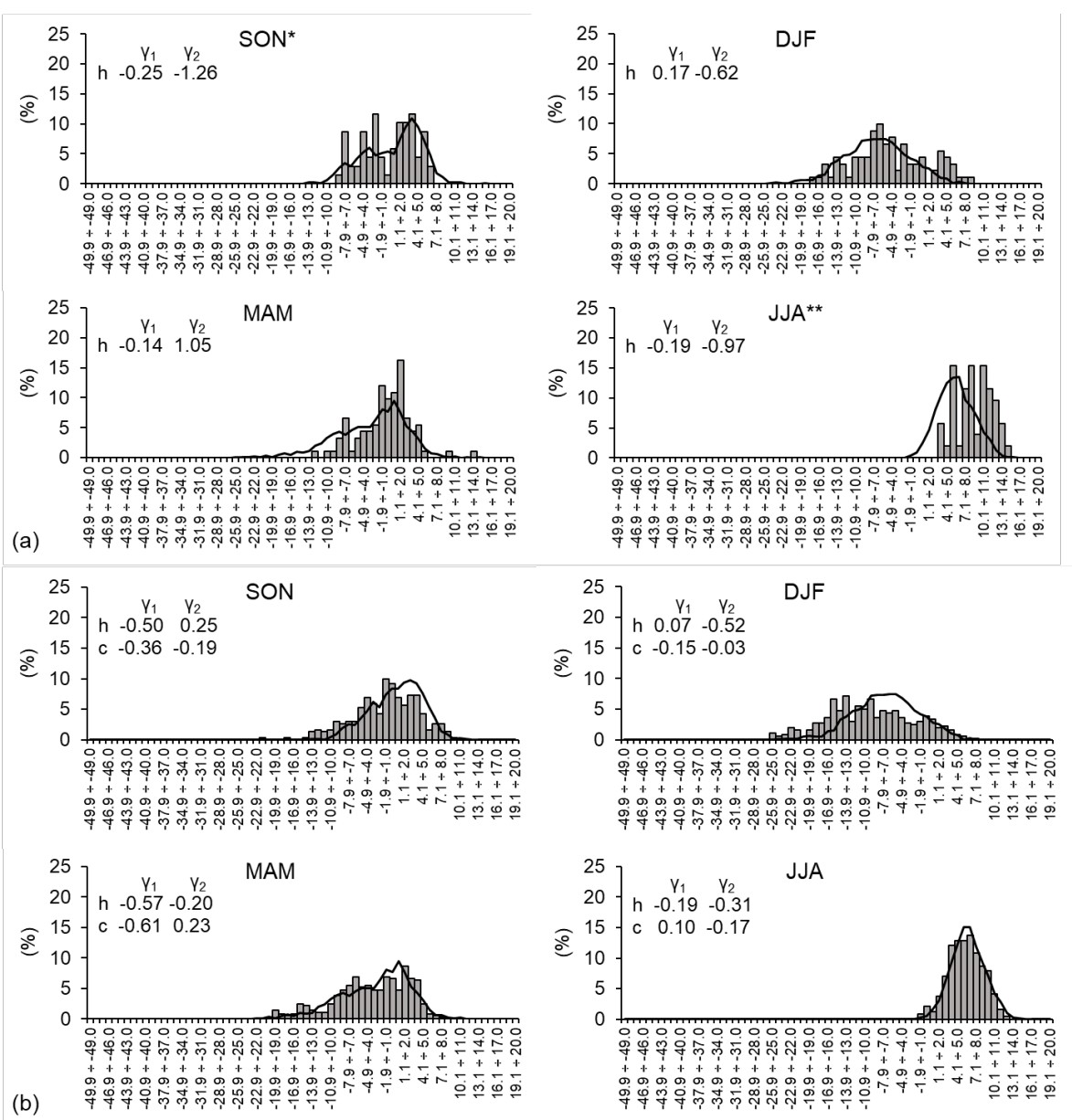

**Figure 8:** Seasonal (SON, DJF, etc.) relative frequencies of occurrence (in %) of MDAT in historical (bars) and modern (lines) sites located in (a) Nuuk 1767–68 and 1991–2020 and (b) Nuuk 1784–92 and 1991–2020. Values of skewness ($\gamma1$) and kurtosis ($\gamma2$) for historical (h) and contemporary (c) times are also shown. Key: * without October; ** for period 1st June to 22nd July

Annual courses of DDTV in Nuuk are shown for each of six historical years for which complete or near-complete data are available for the entire year (Fig. 9). Similarly to the present climate (Przybylak, 2002b), the DDTV in those years was greatest in winter (in particular Dec–Mar, sometimes until Apr) and lowest in the second half of spring, autumn, and

especially summer (Fig. 9). In summer, values of DDTV in all historical years are approximately similar, usually not exceeding 2.0–3.0 °C. In winter, DDTV is on average 2–3 times greater than in summer. In each year there are some periods in which DDTV exceeds 10 °C, reaching maximally about 14 °C (Fig. 9).

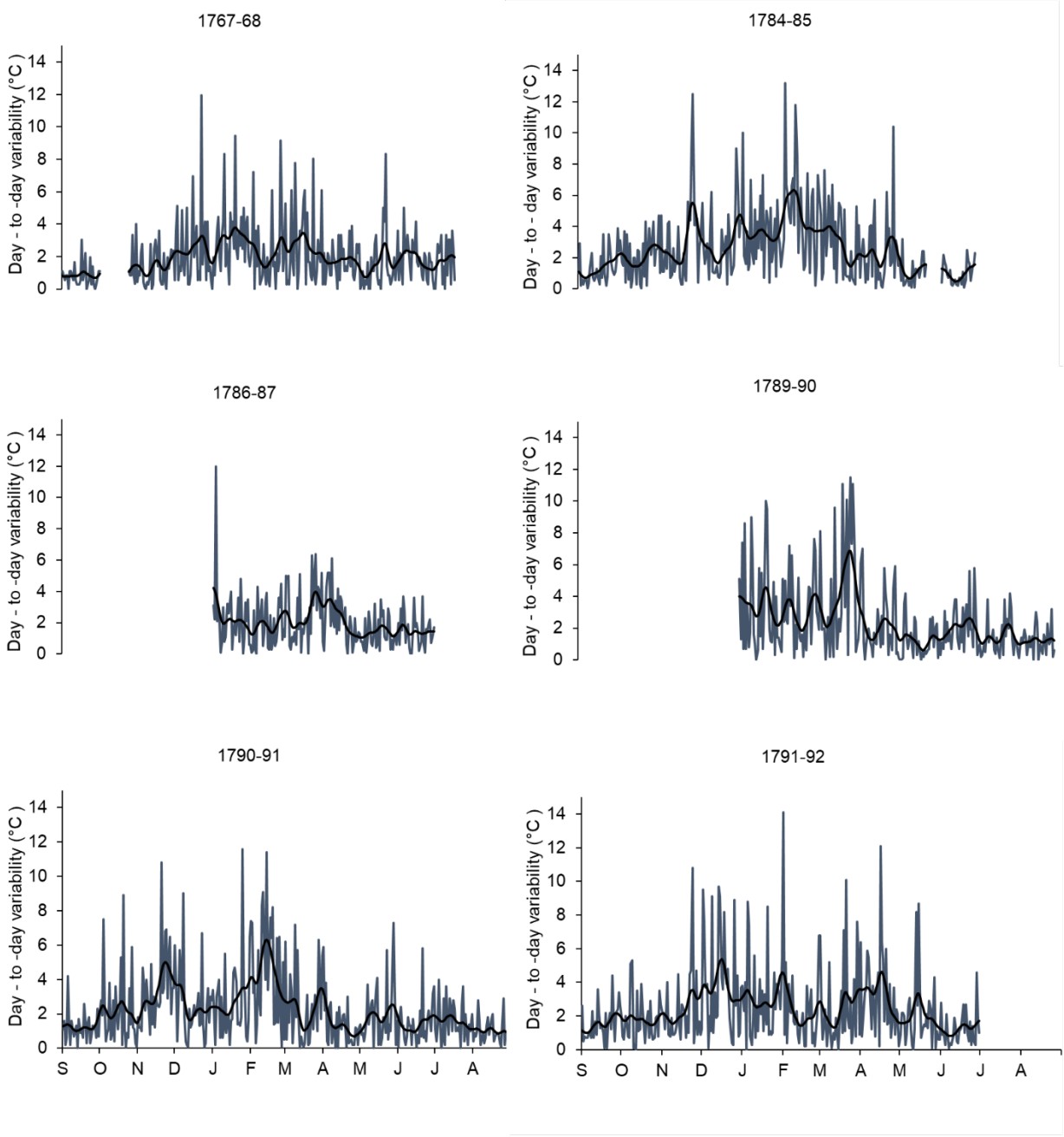

**Figure 9:** Annual courses of DDTV in Nuuk in historical periods. Individual days (grey) are filtered by a Gaussian low-pass filter (black) with a standard deviation of three days in its distribution, corresponding to a rectangular filter of about 10 days.

The DDTV in every studied historical year was usually greater than in the two thermally contrasted years chosen

from the contemporary period (1992/93 [cold year], 2018/19 [warm year]) but only in autumn and winter, whereas DDTV was

smaller in spring and summer (Fig. 10, Fig. S1). This is closely connected with the observed colder-than-present conditions in

autumn and winter, and the absent or near-absent change in thermal conditions in spring and summer. The DDTV differences

rarely exceed ±4 °C. It is important to note, however, that the extreme positive DDTV differences (exceeding 10 °C) were

about two times greater than the negative ones (which rarely fell below -5 °C) (Fig. 10).

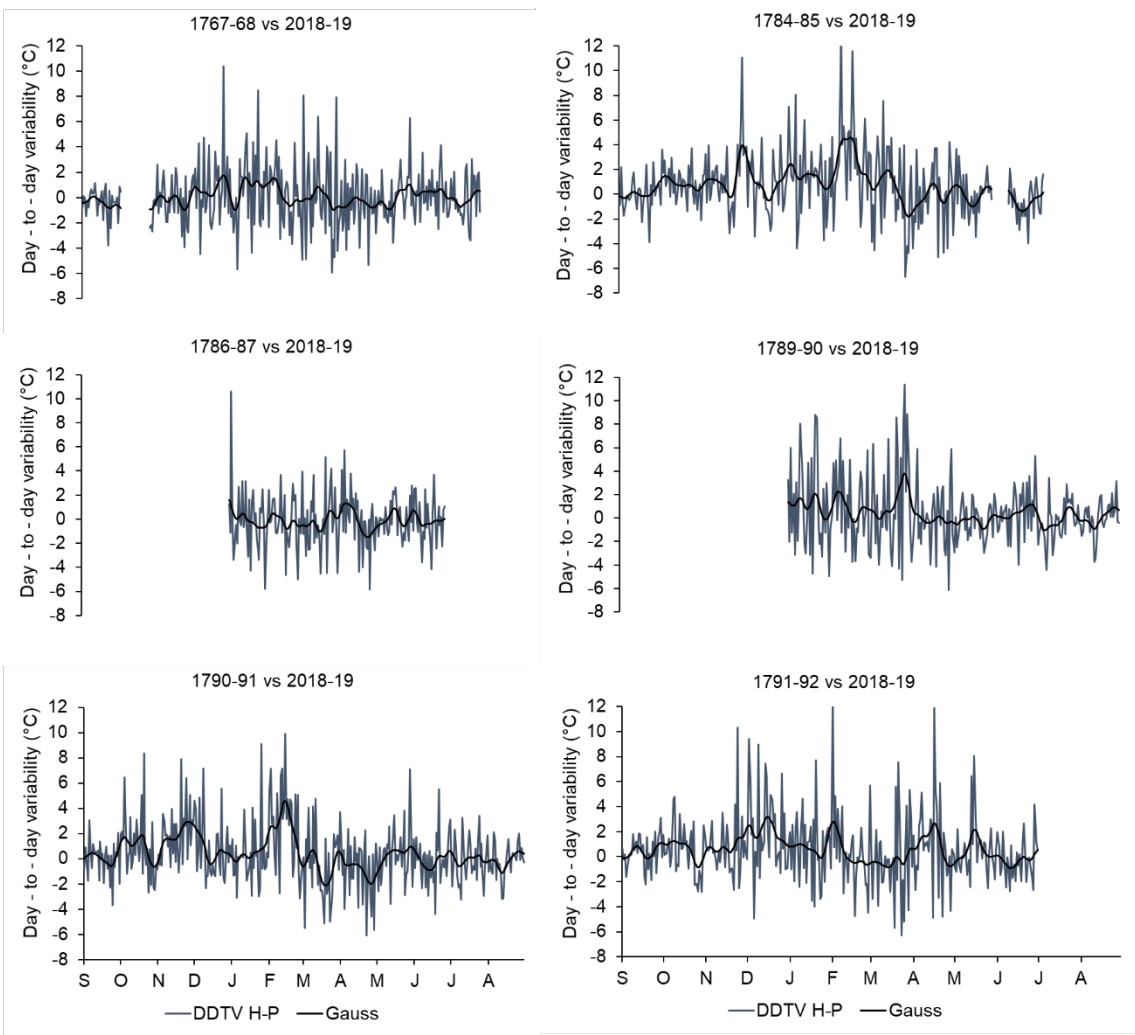


**Figure 10:** Differences in DDTV in Nuuk between historical and contemporary (2018–19) periods. Individual daily differences (grey) are filtered by a Gaussian low-pass filter (black) with a standard deviation of three days in its distribution, corresponding to a rectangular filter of about 10 days.

In the majority of climatological studies (including ours), the year is usually divided into four seasons (DJF, MAM,

etc.). For moderate latitudes, such a division makes physical sense (it more or less captures the annual cycle). It is also

convenient for comparison purposes. In the Arctic, however, the annual cycle is significantly more flat and less clear than at moderate latitudes and is dominated by the winter, which is much longer than the other seasons (Przybylak 2016). Another major weakness of the arbitrarily defined seasons is that we lose information about the season's onset, end and duration (Baranowski 1968). Such knowledge is very useful and important to, for example, the development of vegetation, undertaking economic activity and the lives of indigenous people. For these reasons, we decided to use the thermal criteria proposed by Baranowski (1968) to delimit four standard seasons. The results of some of our investigations are presented in Fig. 11.

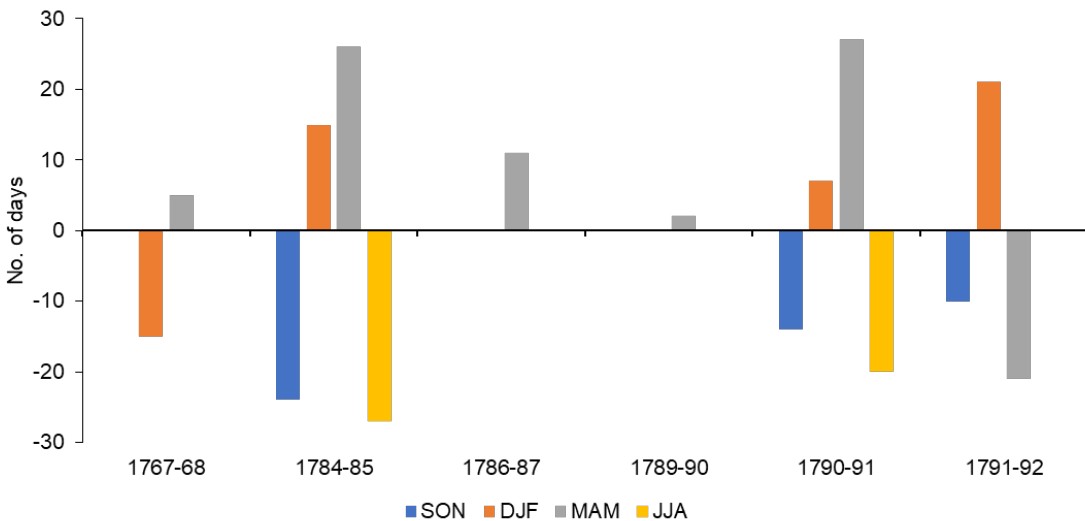

**Figure 11:** Changes in duration of thermal seasons in Nuuk between historical and contemporary (1991–2020) periods (contemporary data were subtracted from historical ones)

The figure clearly shows that the 18th-century Arctic was characterised by the longer springs and winters (except winter of 1767/68) of even up to 20–25 days, while autumns and summers were 10–25 days shorter (except spring of 1791).

## 4 Discussion

A long-term perspective on the Arctic climate and its changes is badly needed, as we wrote in the Introduction section, including to validate the quality of the reconstructions based on proxy data and simulations that are produced by climatic models. Good spatial coverage by instrumental meteorological data for the Arctic has only existed since the 1950s (Przybylak 2000), and therefore any older instrumental measurements that exist are crucial for receiving better insight into the character of climate change and variability of the Arctic in the past. Here, we present an estimation of thermal conditions in SW Greenland based on a unique datasets – the longest and oldest sub-daily temperatures available for the second half of the 18th century. According to most recent reconstructions of mean summer or annual air temperatures in the Arctic (Overpeck et al.

1997; Kaufman et al. 2009; Hanhijarvi et al. 2013; McKay and Kaufman 2014; Werner et al. 2018), this was the warmest period within the longer period of about 1600 to 1900, as it was in the Northern Hemisphere (e.g., Moberg et al. 2005; Hegerl

et al. 2007). This warm period, which was the last such episode before the Contemporary Warming Period (CWP), was caused by both the high solar forcing and the low volcanic forcing that have been observed for this time (Overpeck et al. 1997; Bertrand et al. 2002). That is why improving knowledge about this period is crucial and can be helpful not only for studies of natural climate variability in the Arctic, but also for modelling the future climate.

         As far as we know, no such detailed and comprehensive analysis of thermal conditions in SW Greenland for the 18th

century conducted based on sub-daily instrumental measurements is available in the scientific literature for any part in the Arctic. Most representative for the study period are years taken from the period 1784–92, because the 1767/68 year is isolated and, more importantly, was extremely warm. But to estimate the possible range of temperatures in the study period, the data from 1767/68 are also very important. The question, however, arises as to whether such high temperature values, which are significantly greater than the mean temperature in the contemporary reference period (1991–2020), could have occurred during

the Little Ice Age (LIA). The lack of data does not allow us to check whether this year was also so warm in other areas of the Arctic, especially in Greenland. But we also have at our disposal brief synthetic descriptions of weather in Neu Herrnhut for each month written by Cranz (1820). Analysis of these descriptions indicates the occurrence of exceptionally mild weather in the study historical year in many months. For example, he wrote that mild weather conditions occurred in the cold half-year in December, March and in particularly in January. About the latter month Cranz wrote: "*This month, which in Germany was*

*colder than in the year 1740, was in Greenland remarkably mild.*" Also, the period from April to July was warm, in particular in June. He summarised June as follows: "*The air of this month was generally mild, excepting some cloudy forenoon: there was almost constant sunshine and agreeable spring weather, which is rare in Greenland.*" If the thermometer was not correctly protected against the sun's rays (at this time, screens were not used), such weather (i.e., very sunny weather) may have caused some warm bias in measurements. Warm biases in June and in July probably were partly also connected with the relocation of

the measurement site to Pissiksarbik, where, according to Cranz (1820), "*… the sun's rays are more powerful*". In addition, we checked if at present times such warm years as the historical year 1767/68 had occurred. Calculations of mean temperature in Nuuk for a comparable period (Sep–May) revealed that in the contemporary period (Sep 1990–May 2020), there were seven years that were warmer than 1767/68 (-2.3 °C). The three warmest of those years were: 2009/10 (-0.96 °C), 2003/04 (-1.78 °C) and 2012/13 (-1.90 °C). In conclusion, we have no grounds to consider the temperature values in 1767/68 as unrealistic,

with the exception of the values in the summer months, which may be a bit too high if we assume that the thermometer was not protected against the sun's rays and due to relocation of station.

         Due to the longer series of observations, more representative data is available (as we mentioned earlier) for the period 1784–92. Briefly summarising the measurements for this period, it can be stated that air temperature in Greenland was lower than today from September to March and slightly warmer in summer (see Table 3, Fig. 7). On average, the mean annual (Sep–

Jun) temperature was colder than today by 1.4 °C. Especially cold conditions occurred, however, in November, January and February (anomalies -2.4 °C, -4.9 °C and -2.1 °C, respectively). We can compare only part of our results against some existing

reconstructions based on proxy data (Overpeck et al. 1997; Kaufman et al. 2009; Kobashi et al. 2010; McKay and Kaufman 2014; Werner et al. 2018; Hörhold et al. 2023) and modelling simulations (Goosse and Renssen 2003; Crespin et al. 2009, 2013; Crespin 2014) because the resolution of those reconstructions is limited only to seasonal and annual means. Looking

also at 29 temperature reconstructions for the last four hundred years from different areas of the Arctic (see Fig. 2 in Overpeck et al. 1997), it is clear that temperature changes in different parts of the Arctic are very often not similar to one another, or even opposite. McBean et al. (2005) summarised this fact as follows: "The Arctic is not homogeneous and neither is its climate, and past climate changes have not been uniform in their characteristics or their effects." Moreover, as Lücke et al. (2021) found, orbital forcing strongly influences the seasonal temperature trends during the millennium. Also, this forcing changes

with latitude, and therefore they suggest using only seasonally homogeneous data for reconstructing multicentennial variability (which is especially important in multiproxy reconstructions). A little earlier, Crespin et al. (2013) also found that mean Arctic temperature displays a decreasing trend during the pre-industrial period, whereas spring temperature appears to rise. They attribute this difference in trends to the variations in the Earth's orbital parameters. Thus, it is obvious that we should take into account these facts in interpreting the results. They may to some extent have hampered the comparison of our data from SW

Greenland against data averaged for the entire Arctic and other parts of the Arctic. Therefore, we focus mainly on reconstructions available for Greenland and surrounding areas. The various climate indices based on MDAT that we present here are, unfortunately, still not available in all the mentioned reconstructions based both on proxy data and modelling simulations, and we cannot therefore make any comparison.

Kobashi et al. (2010) reconstructed the history of surface air temperature in central Greenland using isotopes of N2

and Ar in air bubbles in an ice core. As we mentioned in the Introduction section, they found a cold period in the second half of the 18th century here – the coldest in the entire millennium's history. Temperature difference in comparison to the warm periods observed in the first half of the 12th century and at present (second half of the 20th century) reaches 1.5–2.0 °C. A slightly smaller change in temperature between the study period relative to present temperatures (difference 1.0–1.5 °C) is also shown in a reconstruction for north and central Greenland recently published by Hörhold et al. (2023). This range of

temperature change is similar to that (1.4 °C) calculated by us based on instrumental measurements (Table 3). The scale of warming that occurred in Greenland in the late-18th century is comparable to that noted in the Medieval Warm Period (MWP) (Hörhold et al. 2023), though this is not observed in the reconstructions available for the entire Arctic (see Fig. 3 in Werner et al. 2019). Hörhold et al. (2023) also found a warm wave, which is clearly seen in comparison to the neighbouring periods (the first halves of the 18th and 19th centuries). On the other hand, the opposite temperature change to the mentioned neighbouring

periods is revealed by a reconstruction made by Kobashi et al. (2010). There are also some other differences between the two millennial reconstructions of temperature for Greenland, e.g. concerning the scales and the times of occurrence of the MWP and LIA. A clear warming in the study period is seen also in some temperature reconstructions from the northern part of continental Canada based on tree-ring widths (see sites 13, 14 and 26 in Fig. 2 in Overpeck et al. 1997). The reconstructions for the entire Arctic (Kaufman et al. 2009; McKay and Kaufman 2014; Werner et al. 2018, and some other shown in Fig. 3 in

this publication), as we mentioned earlier, also show warming in the second half of the 18th century, but it is smaller than in Greenland (Hörhold et al. 2023).

The modelling reconstruction of the Arctic (defined as the area above 70° N) temperature for the study period (Goose and Renssen 2003) shows results more similar to the reconstructions shown by Werner et al. (2019) and Hörhold et al. (2023). However, the warm wave is longer and encompasses almost the entire 18th century, excluding the last decade. Another difference is that, according to the models' simulation, the first half of the 18th century was warmer than the second, which is not observed in the reconstructions based on proxies. Newer results presented by Crespin et al. (2009) and Crespin (2014) show that the average Arctic (64–80° N) temperature simulated by five models performed with data assimilation agree significantly better with temperature reconstructions based on proxy data (see Fig. 5b in Crespin et al. 2009). In this model simulation, the second half of the 18th century is evidently warmer than the first half. It is also important to underline the existence of the great stability of climate in this time. This means that the limited number of years with instrumental data available to us for the second half of the 18th century can quite well represent the entire warm period.

The differences that exist between the discussed reconstructions are the result of differences in reconstruction methods, quality of proxies, ways of calibrations, areas, etc. We hope that data rescue activity, which has recently been significantly intensified (Brönnimann et al. 2019; Lundstad et al. 2023) and which also included the gathering of our data, should help to eliminate or reduce some of these differences thanks to the potential for improved calibration of reconstructions and to the greater density of instrumental data. It is obvious that quality-controlled and corrected measurement meteorological data are more reliable for describing the climate and its variability in the Arctic (and any other region) than temperature reconstructions based on proxy data.

A more detail comparison using some climate indices calculated for Greenland is possible only for the Early Twentieth Century Arctic Warming Period (ETCAW), for which we have analogous calculations done for the Ilulissat station (Przybylak et al. 2022), which is located quite close to Nuuk. The ETCAW period (1921–50) in north and central Greenland had the same temperature as the period 1784–92. The calculated temperature anomalies in relation to the 1961–90 reference period using data available in Hörhold et al. (2023) were equal to 0.35 and 0.36 °C, respectively (see also Fig. 1 in Hörhold et al. 2023). A similar scale of warming also occurred in some isolated periods during the MWP; for example, in the period 1021–50 there was a temperature anomaly of 0.27 °C. However, when we take the set of 10 years (1939–46, 1948 and 1950) used by Przybylak et al. (2022) to calculate statistics of climate indices for ETCAW based on MDAT from Ilulissat, it turned out that the temperature in these years was on average 0.5 °C colder than in the period 1784–92.

Several papers based on climate model simulations argue that high-frequency temperature variability should decrease in a warmer climate (e.g., Houghton et al. 1990, 1992, 1996; Karl et al. 1995, and references cited therein; Mearns et al. 1995; Zwiers and Kharin 1998; Moberg et al. 2000; Screen 2014). Thus, theoretically, we should observe a smaller DDTV in the study period than in the ETCAW period. Comparison of the results (Table 3 in this paper and Table 4 in Przybylak 2022), however, reveals the opposite relation in the Arctic. The DDTV in both comparable periods was the same in spring (2.2 °C) and autumn (1.8 °C), while in summer it differed by only 0.1 °C. The biggest difference was noted in winter (0.6 °C), when

the DDTV was greater in the historical time (3.2 °C) than in the ETCAW period (2.6 °C). So too in relation to the modern
period (1991–2020), the DDTV was greater in the historical time in every season except summer, when no change was noted
(Table 3). Thus, in this case, these results support the finding that, in a warmer climate, the DDTV is smaller.

Analysis of the duration of seasons in Ilulissat (ETCAW) and in Nuuk (the historical period) confirmed that the
ETCAW period was colder than the study period, especially in winter. Winter was 22 days longer in the ETCAW (204 days)
than in the study period (182), and even 40 days longer than the present day (1991–2020). The length of summers between all
these three warm periods in the Arctic in the last 250 years differs significantly less – from 122 days at present to 111 during
the ETCAW and 107 in the 18th century. As some kind of compensation for winters being shorter in the study period than in
the ETCAW period, we can assume springs were more than twice as long (49 days versus 23 days). No change in duration
was noted for the autumn (27 days in both periods).

Summarising the discussion, we can conclude that the climate was more continental in the study period than at present,
mainly due to lower winter temperatures. It is interesting that summers in all three of the warm periods that occurred in SW
Greenland in the last 250 years (historical period, ETCAW, CWP) showed very similar conditions. The majority of
reconstructions based on proxy data and simulation of climate using climatic models also confirmed that the second half of the
18th century was warm. Thus we can assume that similar environmental changes (recession of glaciers, degradation of
permafrost, sea-ice reduction, etc.) as those we observe today, but at a slightly smaller scale, may also have occurred in the
historical period. The climate conditions were also favourable for the development of plants and for human activity, including
that of the Moravian Missionaries, who built settlements in this area. The biometeorological and bioclimatological conditions
will be analysed in detail in our future paper that is in preparation.

## 5 Conclusions and final remarks

The main results of the present paper can be summarised as follows:

**1.** Compared to present day (1991–2020), air temperature in Nuuk was on average warmer in 1767–68 and colder in 1784–92.
In 1767–68, the turn of December to January was exceptionally warm, with positive MDAT reaching even 5 °C. Summer, too,
was significantly warmer than today. On the other hand, in 1784–92, autumn and particularly winter were markedly colder
than today, while temperatures in the rest of the year were usually slightly warmer than at present (Table 3, Fig. 4).

**2.** The expedition year 1767/68 was warm, and therefore GDD and ATI for each month were equal to or higher than the norm
observed in 1991–2020 (Fig 5a, b). No important changes were observed for PDD, while AFI was usually lower than the
present-day norm (Fig. 5c, d).

**3.** The GDD and ATI in the period 1784–92 usually (except 1787) do not exceed the maximum and minimum values from
1991–2020 (Fig. 6a, b). The intensity of warm events (PDD) during the cold season (from October to April) in the period
1784–92 is close to the average and minimum PDD in 1991–2020, but the AFI values in 1784–92 are between the average and
maximum AFI values calculated for 1991–2020 (Fig. 6 c, d).

**4.** MDAT in historical periods rarely exceed values of ±2 SD of the long-term mean calculated for the contemporary period (Fig. 7).

**5.** The distribution of MDAT was usually close to normal, both in historical and contemporary periods (Fig. 8).

**6.** No change in the annual course of the DDTV in comparison to the present time was found (Table 3, Fig. 9). The DDTV in the historical and ETCAW periods was the same in spring (2.2 °C) and autumn (1.8 °C), while in summer it differed by only 0.1 °C. The biggest difference was noted in winter (0.6 °C), when the DDTV was greater in the historical time (3.2 °C) than in the ETCAW period (2.6 °C). So too in comparison to the modern period (1991–2020), the DDTV of the historical period was greater in every season except summer, when no change was noted (Table 3, Fig. 10).

**7.** The studied historical period appears to have been warmer than the ETCAW period but colder than today, in terms of both mean values and lengths of seasons (Fig. 11).

8. The climate was more continental in the study period than at present, mainly due to the lower winter temperatures observed in the first period. The summer temperatures being very similar did not influence the difference in degree of climate continentality.

9. The relatively warm climate that occurred in the late-18th century probably significantly impacted environmental changes similar to those observed in present time and during the ETCAW, and was favourable to the development of plants and human activity.

Finally, we are obliged to underline that the presented results may still incorporate some biases. We corrected and eliminated the influence of differences in times of day at which measurements were taken had on the MDAT, but we were not able to correct biases connected with the exposition of the thermometers, which are unknown, but which could have influenced the measurements, mainly during the polar day. One positive note is that the low solar elevation at high latitudes should reduce this bias, but on the other hand the presence of the sun's rays during 24 hours in a polar day may have introduced some additional biases if there was no protection of thermometers against the sun's rays (here, locating the thermometer on a north wall is not a solution). Definitely, the smallest biases in the polar latitudes are observed in the cold half-year, and particularly during the polar night. Another source of biases is connected with the accuracy of the thermometers used, which we also were not able to eliminate. But as we have already mentioned, even with such sources of potential errors, the presented instrumental data are of better quality than the data obtained from reconstructions using proxy data or simulated by models. Moreover, palaeoclimatic reconstructions hardly produce seasonal (mainly summer) and annual means and at present are not able to give any reliable information about daily and sub-daily data.

A discover of new instrumental meteorological data, surely still available in archives and libraries, is absolutely crucial not only to improve our knowledge of climate and climate variability in the Arctic, but also in order to better recognise the "workings" of the Arctic climatic system in historical times (which in turn can significantly help in the attribution of causes of observed changes). Such knowledge is also badly needed to better calibrate currently available and future proxy data and to help validate the climate simulations made by numerical models.

**Author contributions**

Study design by RP, AA and PW. Data collection and selection by AA, PW, and RP. Data curation by GS and KCh. Literature review by RP. Statistical analysis and visualization by GS, KCh, PW and RP. Interpretation of results by RP, GS, PW, and AA. Preparation of manuscript by RP with contributions from all co-authors.

**Financial support**

The work was supported by the National Science Centre, Poland project No. 2020/39/B/ST10/00653.

**Availability statement**

Datasets for this research were derived from the following public domain resources:

**1)** Repository for Open Data (RepOD), Nicolaus Copernicus University Centre for Climate Change Research collection, https://repod.icm.edu.pl/dataverse/ncu-cccr, as cited in Singh et al. (2023)

**2)** Danish Meteorological Institute (DMI), https://www.dmi.dk/publikationer/ as cited in Jensen (2022)

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

Cranz, D.: Historie von Grönland: enthaltend die Beschreibung des Landes und der Einwohner &c. insbesondere die Geschichte der dortigen Mission der Evangelischen Brüder zu Neu-Herrnhut und Lichtenfels. Barby: bey H.D. Ebers., 1765.

Crantz, D.: The History of Greenland: Containing A Description of the Country, and Its Inhabitants: and particularly, A Relation of the Mission, carried on for above these Thirty Years by the Unitas Fratrum. 2 vols., London: Printed for the Brethren's Society for the Furtherance of the Gospel among the Heathen., 1767.

Cranz, D.: Fortsetzung der Historie von Grönland, insonderheit der Missions-Geschichte der Evangelischen Brüder zu Neu-Herrnhut und Lichtenfels von 1763. bis 1768. nebst beträchtlichen Zusätzen und Anmerkungen zur natürlichen Geschichte. Barby: bey H.D. Ebers, 1770.

Crantz, D.: The history of Greenland including an account of the mission carried on by the United Brethren in that country. From the German of Dawid Crantz. With contribution to the present time; illustrative notes; and an appendix, containing a sketch of the mission of the Brethren in Labrador, vol 1 and 2, London: Printed for Longman, Hurst, Rees, Orme, and Brown, Paternoster-Row., 1820.

Crespin, E.: Arctic climate variability during the past millennium: combining model simulations and proxy data, Ph.D. thesis, Univeristé Catholique de Louvain, Faculté des Sciences, Ecole de Physique, Earth and Life Institute, Georges Lemaître Centre for Earth and Climate Research, http://hdl.handle.net/2078.1/150597, 135 pp., 2014.

Crespin, E., Goosse, H., Fichefet, T., Mairesse, A., and Sallaz-Damaz, Y.: Arctic climate over the past millennium: Annual and seasonal responses to external forcings, The Holocene, 23(3), 321–329, https://doi.org/10.1177/0959683612463095, 2013.

Crespin, E., Goosse, H., Fichefet, T., and Mann, M.E.: The 15th century Arctic warming in coupled model simulations with data assimilation, Clim. Past, 5, 389–401, https://doi.org/10.5194/cp-5-389-2009, 2009.

Demarée, G.R., and Ogilvie, A.E.J.: The Moravian missionaries at the Labrador coast and their centuries-long contribution to instrumental meteorological observations, Clim. Change, 91, 423–450, https://doi.org/10.1007/s10584-008-9420-2, 2008.

Demarée, G.R., and Ogilvie, A.E.J.: Missionary Activity in Greenland and in Labrador: The Context for Early Meteorological Observations, in: Legacies of David Cranz's 'Historie von Grönland' (1765) edited by Jensz, F., and Petterson, C., Christianities in the Trans-Atlantic World, Springer, 141-164, https://doi.org/10.1007/978-3-030-63998-3_7, 2021.

Demarée, G.R., Ogilvie, A.E.J., and Mailier, P.: Early meteorological observations in Greenland and Labrador in the 18th century: a contribution of the Moravian Brethren in: Proceedings of the 35th International Symposium on the Okhotsk Sea and Polar Oceans (2020), Mombetsu-2020 Symposium, 16–19 February 2020, Okhotsk Sea and Polar Oceans Research Association (OSPORA), Mombetsu, Hokkaido, Japan: 35–38, 2020.

Ginge, A.: Observationes Gotthaabenses Autore Andrea Ginge, pastroe loci. in: Epheremides Societatis Meteorologicae Palatinae, Mannheim: Ex Officina Novae Sociatatis Typographicae, pp. 42–69, 1789.

Goosse, H., and Renssen, H.: Simulating the evolution of the Arctic climate during the last millennium, in: Proceedings Seventh Conference on Polar Meteorology and Oceanography and Joint Symposium on High-Latitude Climate Variations American Meteorological Society. Contribution 1.5, Gebeurtenis: Seventh Conference om Polar Meteorology and Oceanography and Joint Symposium on High-Latitude Climate Variations, 12–16 May, 2003.

Hanhijärvi, S., Tingley, M.P., and Korhola, A.: Pairwise comparisons to reconstruct mean temperature in the Arctic Atlantic
Region over the last 2000 years, Clim. Dyn., 41, 2039–2060. https://doi.org/10.1007/s00382-013-1701-4, 2013.

Hegerl, G.C., Crowley, T.J., Allen, M., Hyde, W.T., Pollack, H.N., Smerdon, J., and Zorita, E.: Detection of human influence on a new, validated 1500-year temperature reconstruction, J. Clim., 20, 650–666 https://doi.org/10.1175/JCLI4011.1, 2007.

Climate Change: The IPCC Scientific Assessment edited by Houghton, J.T., Jenkins, G.J., and Ephraums, J.J., Cambridge University Press, Cambridge, 365 pp., 1990.

Drost Jensen, C. (Ed.): Weather Observations from Greenland 1958-2021. DMI Report 22-08. Danish Meteorological Institute https://www.dmi.dk/publikationer/, 2022.

Houghton, J.T., Jenkins, G.J., and Ephraums, J.J. (Eds.): Climate Change: The IPCC Scientific Assessment. Cambridge University Press, Cambridge, 365 pp., 1990.

Houghton, J.T., Callander, B.A., and Varney, S.K. (Eds.): Climate Change: The Supplementary Report to the IPCC Scientific
Assessment, Cambridge University Press, Cambridge, 200 pp., 1992.

Houghton, J.T., Meila Filho, L.G., Callander, B.A., Harris, N., Kattenberg, A., and Maskell, K. (Eds.): Climate Change 1995: The Science of Climate Change. Cambridge University Press: Cambridge; 572 pp., 1996.

Hörhold, M., Münch, T., Weißbach, S., Kipfstuhl, S., Freitag, J., Sasgen, I., Lohmann, G., Vinther, B., and Laepple, T.: Modern temperatures in central–north Greenland warmest in past millennium, Nature, 613, 503–507,
https://doi.org/10.1038/s41586-022-05517-z, 2023.

Huybrechts, P. and Oerlemans, J.: Response of the Antarctic Ice Sheet to future greenhouse warming. Climate Dynamics 5, 93–102, 1990.

Instanes, A.: Incorporating climate warming scenarios in coastal permafrost engineering design—case studies from Svalbard and northwest Russia. Cold Regions Science and Technology 131, 76–87, doi: 10.1016/j.coldregions.2016.09.004, 2016.

IPCC, 2021: Climate Change 2021: The Physical Science Basis. Contribution of Working Group I to the Sixth Assessment Report of the Intergovernmental Panel on Climate Change edited by Masson-Delmotte, V., Zhai, P., Pirani, A., Connors, S.L., Péan, C., Berger, S., Caud, N., Chen, Y., Goldfarb, L., Gomis, M.I., Huang, M., Leitzell, K., Lonnoy, E., Matthews, J.B.R., Maycock, T.K., Waterfield, T., Yelekçi, O., Yu, R., and Zhou B., Cambridge University Press, Cambridge, United Kingdom and New York, NY, USA, 2391 pp. https://doi.org/10.1017/9781009157896, 2021.

Karl, T.R., Knight, R.W., and Plummer, N.: Trends in high-frequency climate variability in the twentieth century, Nature, 377, 217–220, https://doi.org/10.1038/377217a0, 1995.

Kaufman, D.S., Schneider, D.P., McKay, N.P., Ammann, C.M., Bradley, R.S., Briffa, K.R., Miller, G.H., Otto-Bliesner, B.L., Overpeck, J.T., and Vinther, B.M., Arctic Lakes 2k Project Members: Recent warming reverses long-term arctic cooling. Science 325: 1236–1239, https://www.science.org/doi/10.1126/science.1173983, 2009.

Kobashi, T., P. Severinghaus, J., Barnola, J-M, Kawamura, K., Carter, T., and Nakaegawa, T.: Persistent multi-decadal Greenland temperature fluctuation through the last millennium, Clim. Change, 100, 733–756, https://doi.org/10.1007/s10584-009-9689-9, 2010.

Kodzik, J.: Moravian Missions in the European Arctic during the Enlightenment: collecting, classifying and communicating knowledge (Greenland, Iceland and Lapland), Arctic&Antarctic, 13, 55–71.
https://iacsi.hi.is/issues/2019_volume_13/3_article_vol_13.pdf, 2019.

Kosiba, A.: O konieczności ujednolicenia skali międzynarodowej podstawowych kryteriów termicznych w klimatologii. Przegląd Geofizyczny, 3(11), 27–31, 1958.

Kottek, M., Grieser, J., Beck, C., Rudolf, B., and Rubel, F.: World Map of the Köppen-Geiger climate classification updated, *Meteorol. Z.*, **15**, 259–263, https://koeppen-geiger.vu-wien.ac.at/present.htm, 2006.

Kratzenstein, Ch. G.: Vær Observationer for Aarene 1767 og 1768 anstillede i Grönland. Brägte i Orden og bekiendtgiorte af C.G. Kratzenstein. Skrifter som udi det Kiøebenhavnske Selskab af Laerdoms og Videnskabers Elskere ere fremlagte og oplaeste i Aarene 1765, 1766, 1767, 1768 og 1769, 373–392, 1770.

Lücke, L.J., Schurer, A.P., Wilson, R., and Hegerl, G.C.: Orbital forcing strongly influences seasonal temperature trends during the last millennium, Geophysical Research Letters, 48, e2020GL088776. https://doi.org/10.1029/2020GL088776, 2021.

Lüdecke, C.: East meets West. Meteorological observations of the Moravians in Greenland and Labrador since the 18th century, History of Meteorology, 2, 123–132. https://journal.meteohistory.org/index.php/hom/article/view/22, 2005.

Lundstad, E., Brugnara, Y., Pappert, D., Kopp, J., Samakinwa, E., Hürzeler, A., Andersson, A., Chimani, B., Cornes, R., Demarée, G., Filipiak, J., Gates, L., Ives, G.L., Jones, J.M., Jourdain, S., Kiss, A., Nicholson, S.E., Przybylak, R., Jones, P., Rousseau, D., Tinz, B., Rodrigo, F.S., Grab, S., Domínguez-Castro, F., Slonosky, V., Cooper, J., Brunet, M., and
Brönnimann, S.: The global historical climate database HCLIM, Sci. Data, 10, 44, https://doi.org/10.1038/s41597-022-01919-w, 2023.

McBean. G., Alekseev. G., Chen. D., Førland. E., Fyfe. J., Groisman. P.Y., King. R., Melling. H., Vose. R., and Whitfield P. H.: Arctic Climate: Past and Present, Arctic Climate Impact Assessment. ACIA Overview report. Cambridge University Press. 1020, pp. 21–60, 2005.

McKay, N.P. and Kaufman, D.S.: An extended Arctic proxy temperature database for the past 2,000 years, Sci. Data, 1, 140026, https://doi.org/10.1038/sdata.2014.26, 2014.

Mearns, L.O., Giorgi, F., McDaniel, L., and Shields, C.: Analysis of variability and diurnal range of daily temperature in a nested regional climate model: comparison with observations and doubled CO2 results, Clim. Dyn., 11, 193–209, https://doi.org/10.1007/BF00215007, 1995.

Mills, W.J.: Exploring Polar Frontiers [2 volumes]: A Historical Encyclopedia [2 volumes]. Bloomsbury Publishing USA, 2003.

Moberg, A., Jones, P.D., Barriendos, M., Bergström, H., Camuffo, D., Cocheo, C., Davies, T.D., Demarée, G., Martin-Vide, J., Maugeri, M., Rodriguez, R., and Verhoeve, T.: Day-to-day variability trends in 160- to 275-year-long European instrumental records, J. Geophys. Res., 105 (D18), 22849-22868, https://doi.org/10.1029/2000JD900300, 2000.

Moberg, A., Sonechki, D.M., Holmgren, K., Datsenko, N.M, and Karlen, W.: Highly variable Northern hemisphere temperatures reconstructed from low- and high-resolution proxy data, Nature, 433, 613–617. https://doi.org/10.1038/nature03265, 2005.

Nordli, Ø., Przybylak, R., Ogilvie, A.E.J., and Isaksen, K.: Long-term temperature trends and variability on Spitsbergen: the extended Svalbard Airport temperature series, 1898-2012, Polar Res., 33, 21349,
http://dx.doi.org/10.3402/polar.v33.21349, 2014.

Nordli, Ø., Wyszyński, P., Gjelten, H.M., Isaksen, K., Łupikasza, E., Niedźwiedź, T., and Przybylak, R.: Revisiting the extended Svalbard Airport monthly temperature series, and the compiled corresponding daily series 1898–2018, Polar Res., 39, 3614, http://dx.doi.org/10.33265/polar.v39.3614, 2020.

Nuttall, M. (Ed.): Encyclopedia of the Arctic. Routledge, 2005.

Overpeck, J., Hughen, K., Hardy, D., Bradley, R., Case, R., Douglas, M., Finney, B., Gajewski, K., Jacoby, G., Jennings, A., Lamoureux, S., Lasca, A., MacDonald, G., Moore, J., Retelle, M., Smith, S., Wolfe, A., and Zielinski, G..: Arctic environmental change of the last four centuries. Science 278(5341): 1251–1256, https://www.science.org/doi/10.1126/science.278.5341.1251, 1997.

Przybylak, R.: Temporal and spatial variation of air temperature over the period of instrumental observations in the Arctic, Int.
J. Climatol., 20, 587–614, https://doi.org/10.1002/(SICI)1097-0088(200005)20:6<587::AID-JOC480>3.0.CO;2-H, 2000.

Przybylak, R.: Variability of Air Temperature and Atmospheric Precipitation During a Period of Instrumental Observation in the Arctic. Kluwer Academic Publishers, Boston-Dordrecht-London, 330 pp., 2002a

Przybylak, R.: Changes in seasonal and annual high-frequency air temperature variability in the Arctic from 1951–1990, Int. J. Climatol., 22, 1017–1032. https://rmets.onlinelibrary.wiley.com/doi/pdfdirect/10.1002/joc.793, 2002b.

Przybylak, R.: The Climate of the Arctic, Second Edition, Springer, 287 pp., https://doi.org/10.1007/978-3-319-21696-6, 2016.

Przybylak R., Svyashchennikov, P.N., Uscka-Kowalkowska, J., and Wyszyński, P.: Solar radiation in the Arctic during the Early Twentieth Century Warming (1921–1950), presenting a compilation of newly available data, J. Clim., 33, 21-37, https://doi.org/10.1175/JCLI-D-20-0257, 2021.

Przybylak, R. and Vízi, Z.: Air temperature changes in the Canadian Arctic from the early instrumental period to modern times,
Int. J. Climatol., 25, 1507–1522, https://doi.org/10.1002/joc.1213, 2005.

Przybylak, R., Vizi, Z., and Wyszyński, P.: Air temperature changes in the Arctic from 1801 to 1920, Int. J. Climatol., 30, 791–812, https://doi.org/10.1002/joc.1918, 2010.

Przybylak, R. and Wyszyński, P.: Air temperature in Novaya Zemlya Archipelago and Vaygach Island from 1832 to 1920 in the light of early instrumental data, Int. J. Climatol., 37, 8, 3491–3508, https://doi.org/10.1002/joc.4934, 2017.

Przybylak, R., Wyszyński, P., Nordli, Ø., and Strzyżewski, T., 2016: Air temperature changes in Svalbard and the surrounding seas from 1865 to 1920, Int. J. Climatol., 36, 2899-2916, https://doi.org/10.1002/joc.4527, 2016.

Przybylak, R., Wyszyński, P., and Araźny, A.: Comparison of Early Twentieth Century Arctic Warming and Contemporary Arctic Warming in the light of daily and sub-daily data, J. Clim., 35, 2269-2290, https://doi.org/10.1175/JCLI-D-21-0162.1, 2022.

Screen, J.A.: Arctic amplification decreases temperature variance in northern mid- to high-latitudes, Nat. Clim. Change, 4, 577–582. https://www.nature.com/articles/nclimate2268, 2014.

Singh, G., Chmist, K., Przybylak, R., Wyszyński, P., and Araźny, A.: Air temperature data for SW Greenland in the second half of the 18th century [data set], RepOD, https://doi.org/10.18150/L1Y21Q, 2023.

van Wijngaarden, W.A.: Arctic temperature trends from the early nineteenth century to the present, Theor. Appl. Climatol.,
122, 567–580, https://doi.org/10.1007/s00704-014-1311-z, 2015.

Vinther, B.M., Andersen, K.K., Jones, P.D., Briffa, K.R., and Cappelen, J.: Extending Greenland temperature records into the late eighteenth century, J. Geophys. Res., 111: D11105, https://doi.org/10.1029/2005JD006810, 2006.

Von Storch, H. and Zwiers, F.W.: Statistical Analysis in Climate Research, Cambridge University Press: Cambridge, UK, 484, https://doi.org/10.1017/CBO9780511612336, 1999.

Werner, J.P., Divine, D.V., Ljungqvist, F.Ch., Nilsen, T., and Francus, P.: Spatio-temporal variability of Arctic summer temperatures over the past 2 millennia, Climate of the Past, 14, 527–557, https://doi.org/10.5194/cp-14-527-2018, 2018.

Zwiers, F.W., and Kharin, V.V.: Changes in extremes of the climate simulated by CCC GCM2 under CO2 doubling, J. Clim., 11, 2200–2222, https://doi.org/10.1175/1520-0442(1998)011<2200:CITEOT>2.0.CO;2, 1998.

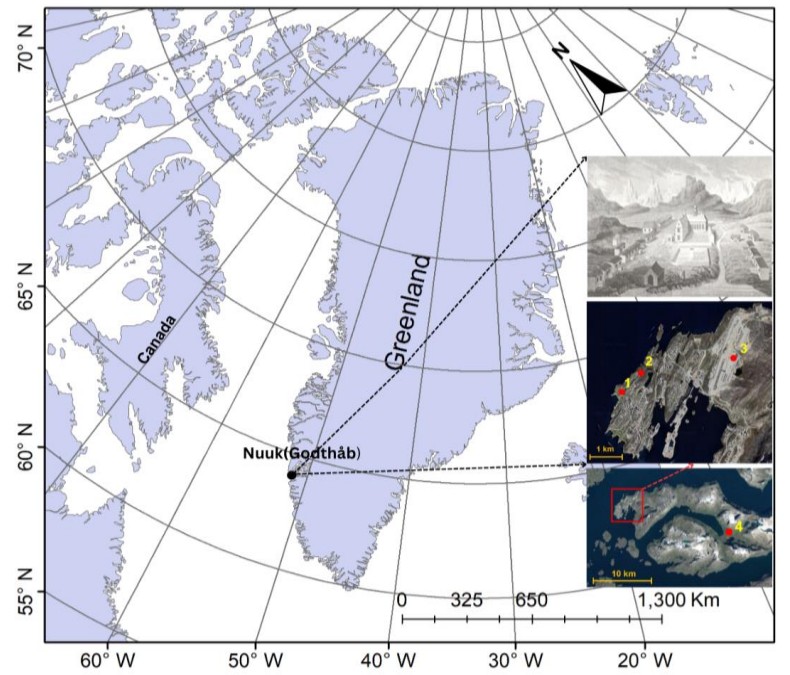

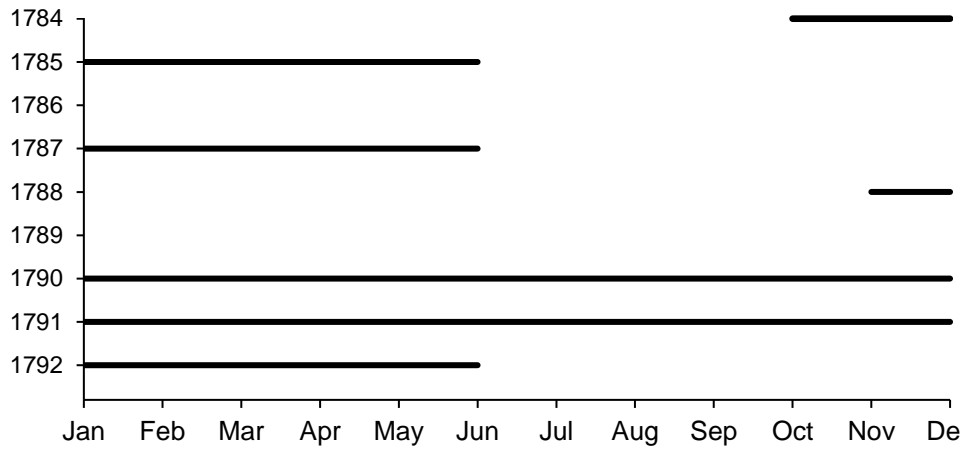

(a) (b)

(a)

(b)

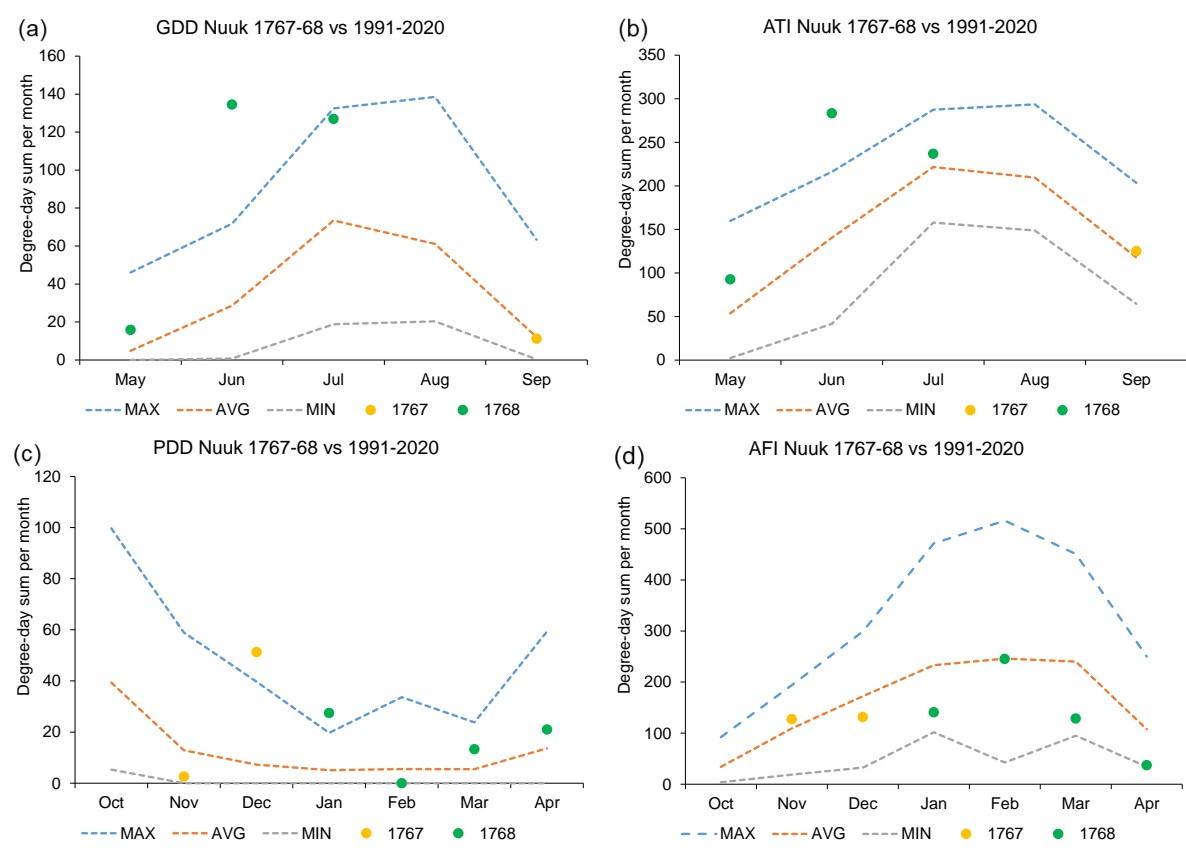

(a) GDD Nuuk 1767-68 vs 1991-2020

(b) ATI Nuuk 1767-68 vs 1991-2020

(c) PDD Nuuk 1767-68 vs 1991-2020

(d) AFI Nuuk 1767-68 vs 1991-2020

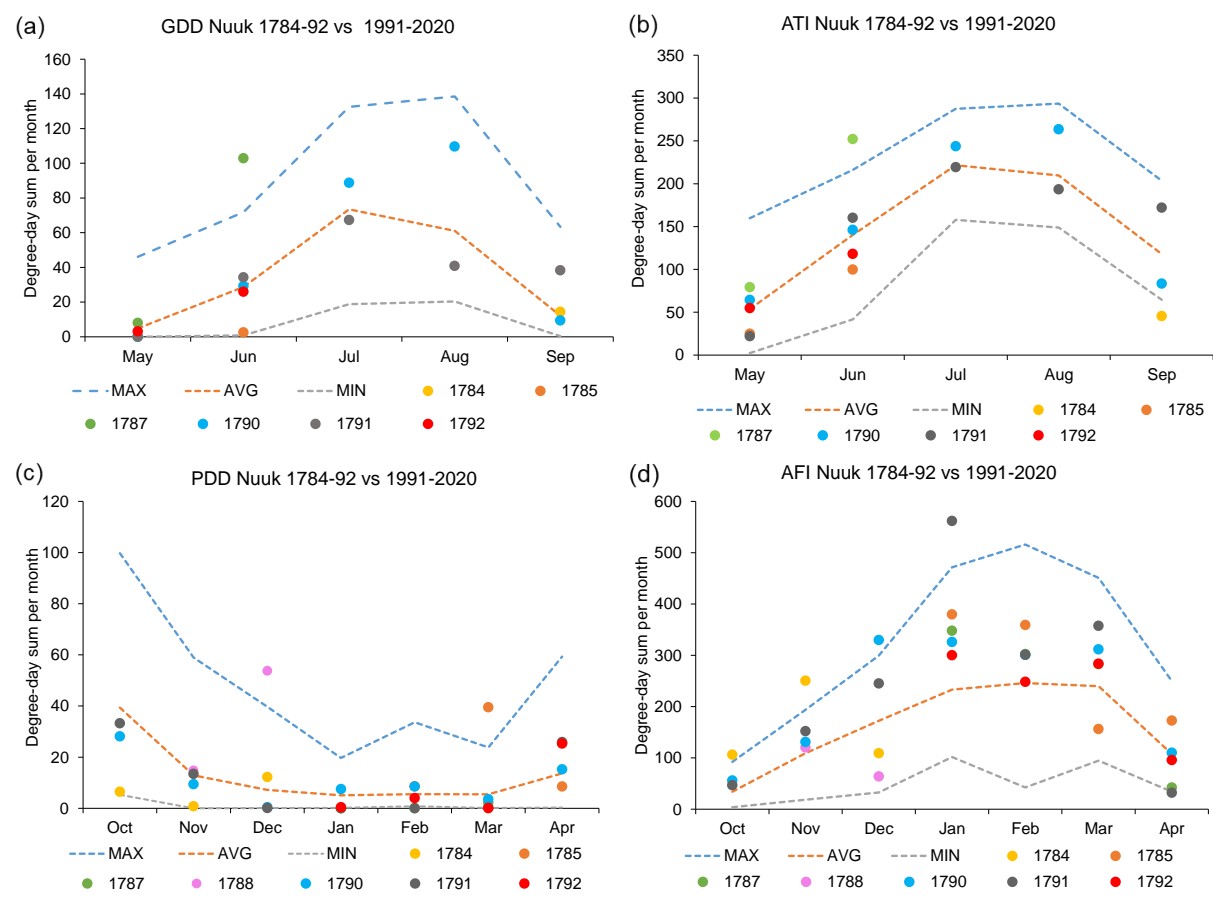

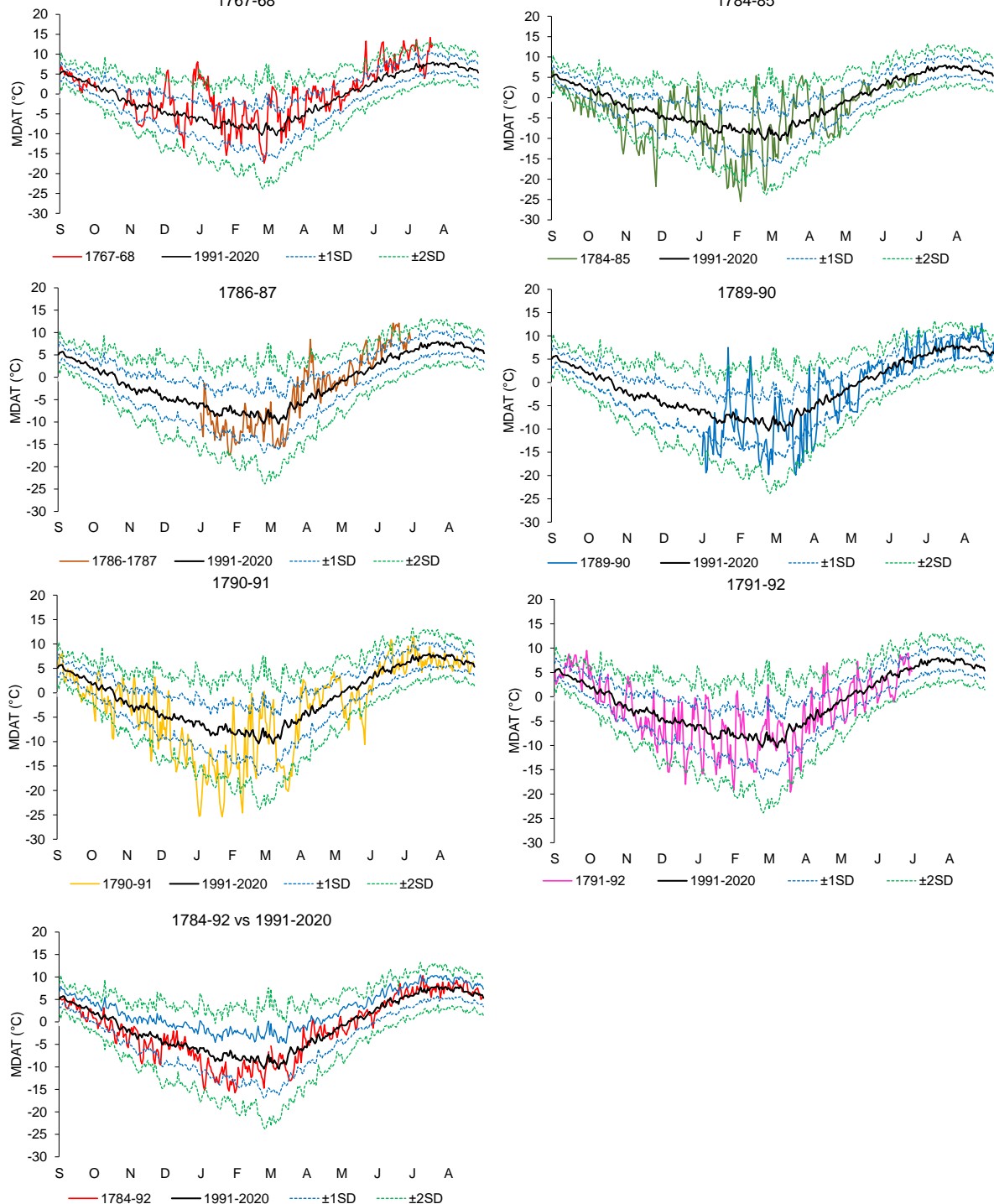

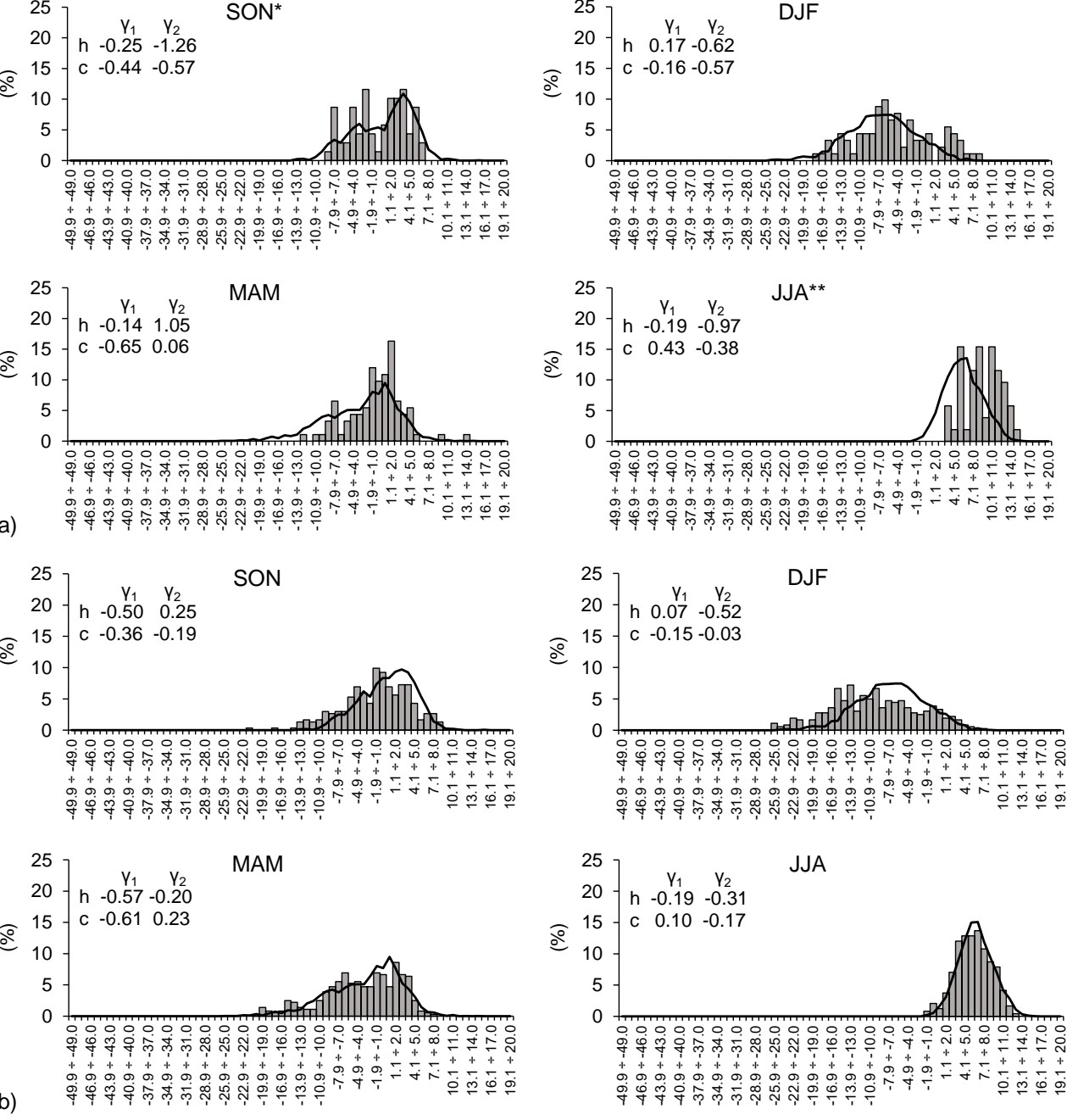

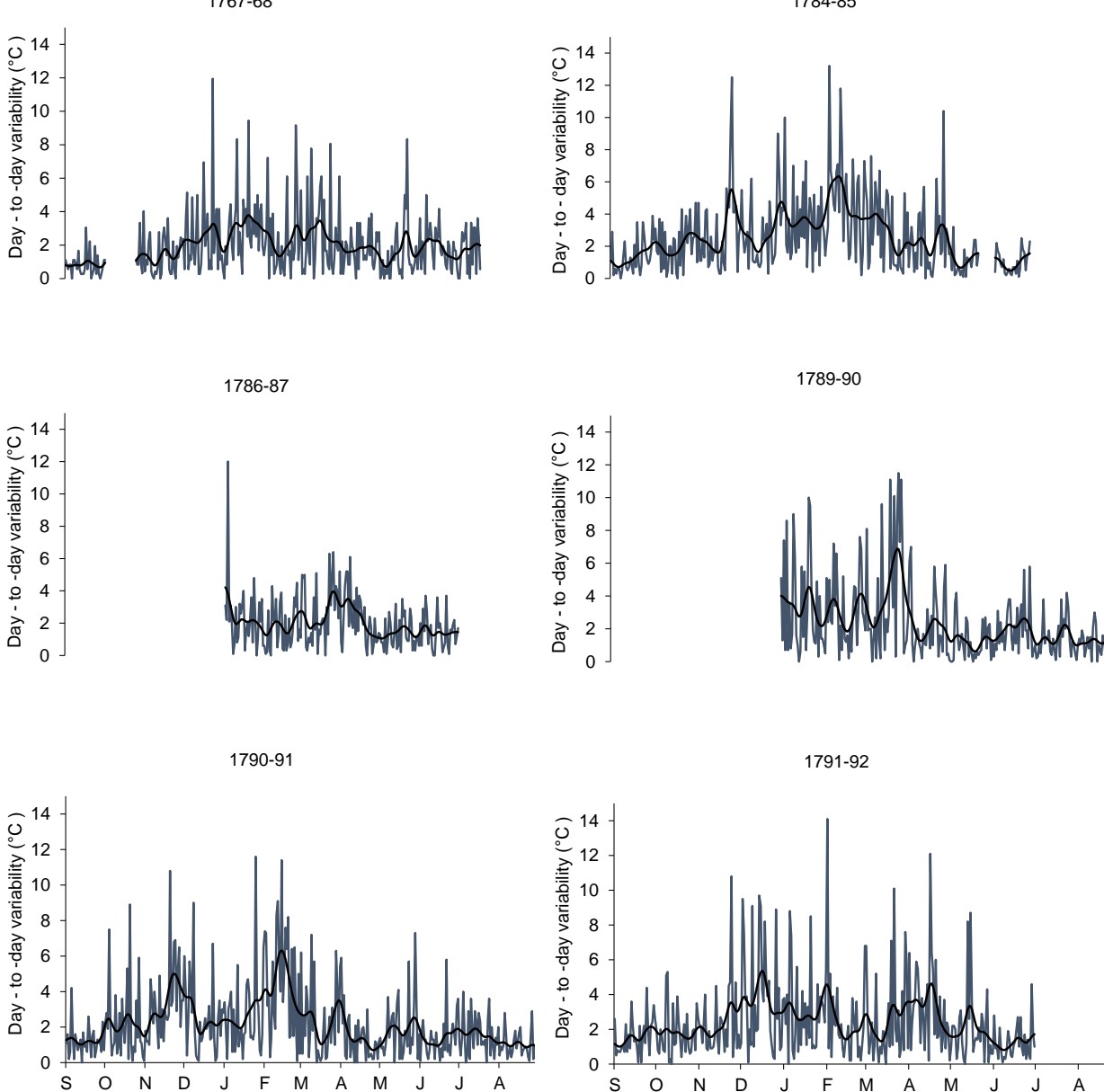

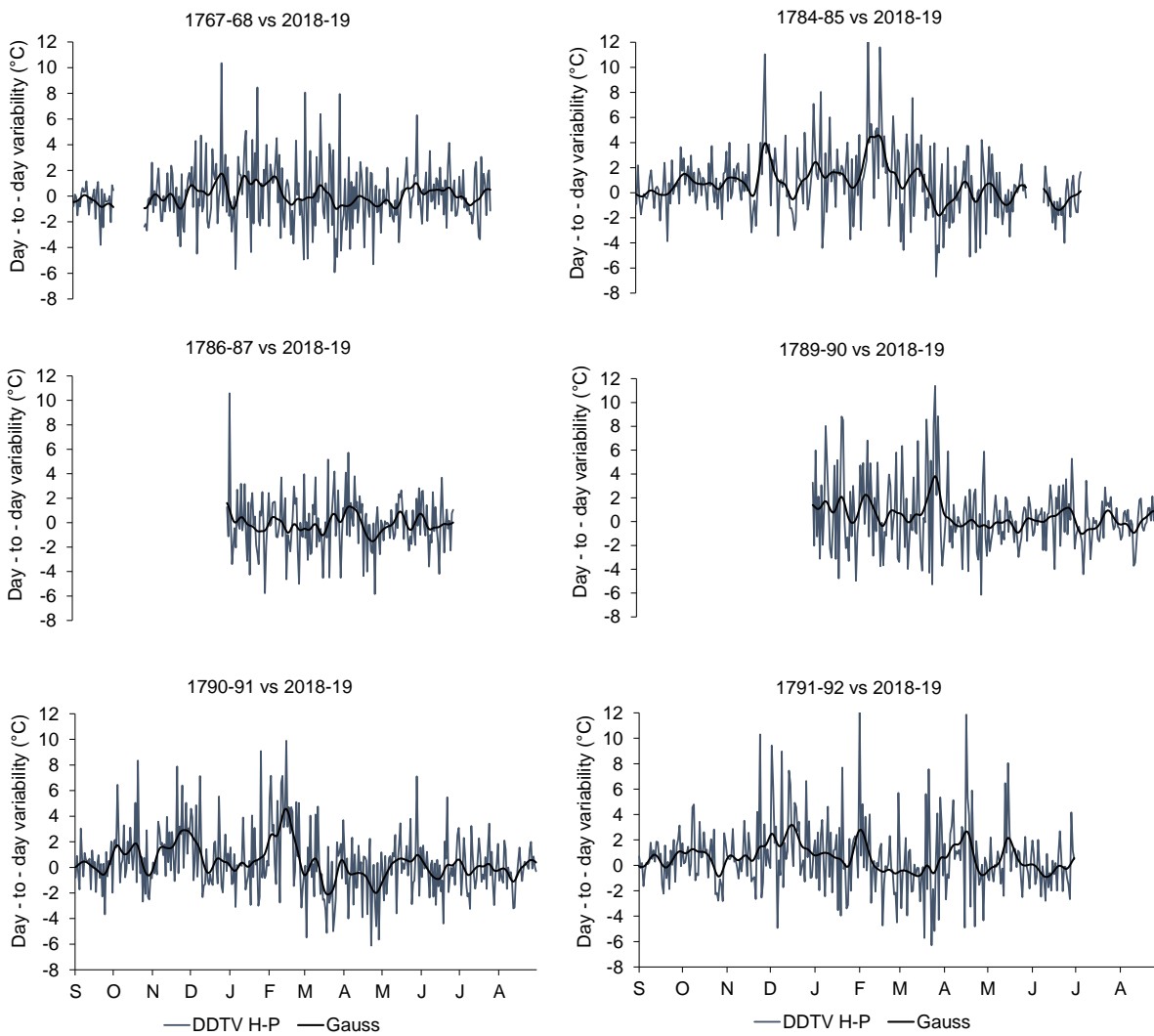

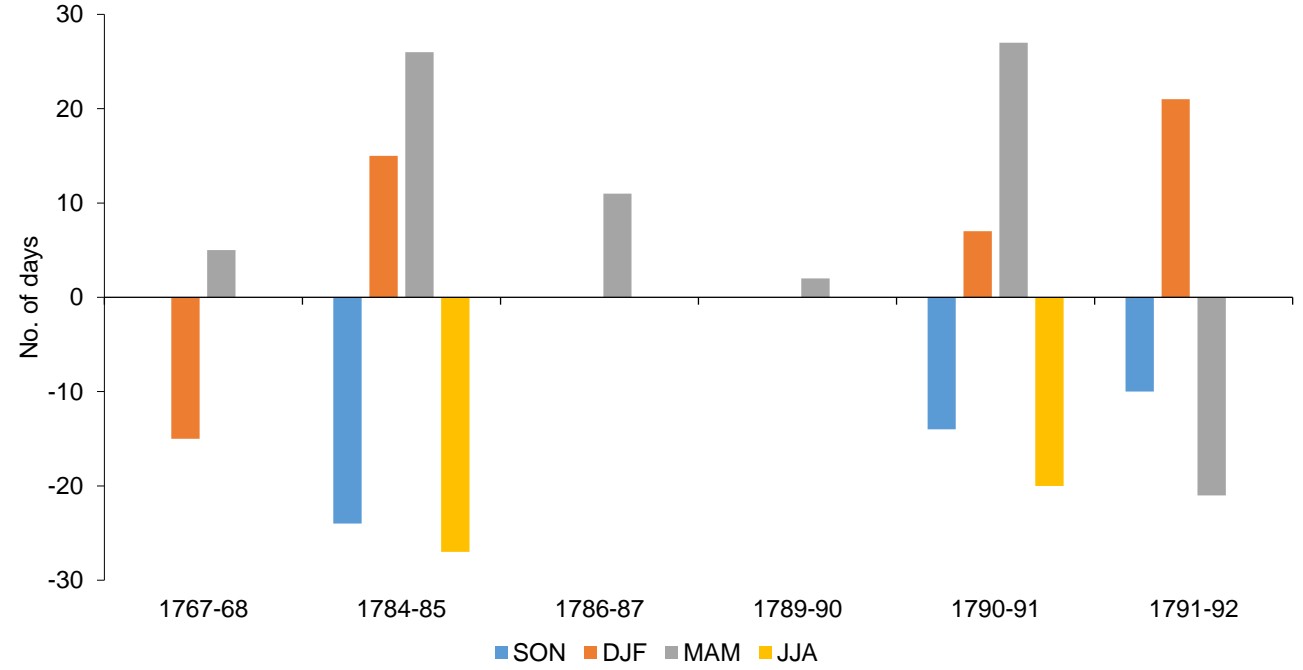