# Peer review of "Air temperature changes in SW Greenland in the second half of the 18th century"

_Climate of the Past, 2023_

## Author Comment (AC1)

Journal: Climate of the Past

Manuscript ID: cp-2023-95

Title: Air temperature changes in SW Greenland in the second half of the 18th century

Dear  Chandal Camendish

Editor, Climate of the Past

We would like to thank you and the anonymous reviewers for providing positive feedback and constructive comments on our manuscript. All comments were carefully considered, and we believe they helped us improve the description of our work. The detailed corrections/modifications are listed below, point by point.

(Note: The changes in the text and the answers to the reviewer's questions/suggestions are marked in red font. We revised the text, taking into account all comments and suggestions proposed by the reviewers. All changes have been carefully applied to the text.)

**Referee No. 1.**

In this work, the sub-daily temperature records of two periods in southwest Greenland, 1767.9.1-1768.7.22h and 1784.9-1792.6 respectively, were used to analyses the climate conditions in the second half of the 18th century. The results were displayed through various statistical and visual methods, combined with some weather and climate indexes. It is important to increase the understanding of the climate in this region of high latitudes at an earlier time. This work has significant implications for increasing the understanding of earlier climate in this region of high latitude. The manuscript has a detailed narrative with rich results and in-depth discussion. However, we still recommend that the readability of the paper be improved through minor revisions and adjustments. Here are a few specific suggestions.

1. Here are 2 minor fallacies in the formulas and charts:

- On page 6, line 139, in equation (4), is there a missing T in front of 21?

ANS: corrected

- In the picture in Figure 7, the letter numbers are in upper case, and in the name of the figure, they are in lower case.

ANS: corrected

2. We suggest a more detailed, systematic and independent description of the sources of the data, the way they are observed, the form in which they are recorded, etc, which are rather scattered and abbreviated in the current text. (Such as the data sources are in the remarks to the figure, the means of observation and missing measurements are briefly mentioned in paragraph 4.) Thus, we recommend a separate subsection in Part 2 to display them, as well as, a basic presentation of the data through statistics and visualization, including time periods of valid data, etc. This will be more visual than a text description. We believe that this will allow readers from different professional backgrounds to be more aware of the observation record and increase the credibility of the analysis results.

ANS: Thank you very much for this suggestion. We have distinguished subsection 2.1 in which we described in detail all available metadata and data. We want to stress again that the presented meteorological data are the oldest that exist for the Arctic, and that in the manuscript we informed

readers about all the metadata that exist. But for greater clarity we added information that the sources give only the time of measurements and the thermometrical scale used. Unfortunately, the information about detailed place of measurements, sheltering and exposure, manufacturers of thermometers, etc. is not available for both series. We also added to the manuscript an additional figure (see below) which presents for the period 1784–92 the availability of data for analysis. This will definitely help readers more easily to estimate the availability of data.

[Figure]

**Figure. Coverage of air temperature observations in Nuuk (orig. Godthaab) in the period 1784–92**

3. How the records were further analyzed could also be a separate section in Part 2. In this section, we think it would have been better to give the reasons for the selection of each climate or weather indices (Table 2). Because it seems that from the results alone, these indices do not give some new and interesting insights other than adding to the richness of the presentation. Therefore, we believe it would be better if the purpose of selecting these indices for analysis was mentioned in the methodology section to help the reader understand the purpose and results, also better echo what would be discussed in part 4.

ANS: Thank you for this comment and suggestion. We have added more information about the indices used and their importance in improving the knowledge of climate in the study area. We think that the presented threshold statistics make the description of climate and weather and their changes in comparison to present conditions more comprehensive and complex and, importantly, very useful for scientists investigating other components of the Arctic Climate System (biosphere, lithosphere, hydrosphere and cryosphere). For example, the GDD index or number of growing days significantly "impacts plants' and animals' activity and growth, which in the Arctic region may start as soon as snow melting has taken place" (Nordli et al. 2020). On the other hand, the ATI index is often used in permafrost engineering, engineering design and for estimations of active layer thickness over the permafrost (Instanes 2016). The PDD index, in turn, is also commonly used by glaciologists, e.g. for modelling glacier or snow melt, which is possible only when the temperature is above 0 °C. The PDD can therefore be thought of, according to Huybrechts and Oerlemans (1990), as the total energy available for melting snow and ice over the course of one year. The temperature oscillation around the 0 °C threshold is also extremely important for studying, e.g., mechanical and chemical weathering processes in the Arctic. Also of importance is the fact that we calculated all of these indices and presented the results in the paper describing temperature change in Svalbard in the period 1898–2018 (see Nordli et al. 2020). Thanks to that, it is possible (and will be possible in future works) to compare the results between the pre-anthropogenic period (in the Arctic *ca* before 1950) and recent warming in both areas (i.e. Greenland and Svalbard).

Huybrechts, P., and Oerlemans, J.: Response of the Antarctic Ice Sheet to future greenhouse warming. Climate Dynamics 5, 93-102, https://doi.org/10.1007/BF00207424, 1990.

Instanes, A.: Incorporating climate warming scenarios in coastal permafrost engineering design—case studies from Svalbard and northwest Russia. Cold Regions Science and Technology 131, 76–87, https://doi.org/10.1016/j.coldregions.2016.09.004, 2016.

Nordli, Ø., Wyszyński, P., Gjelten, H. M., Isaksen, K., Łupikasza, E., Niedźwiedź, T., and Przybylak, R.: Revisiting the extended Svalbard Airport monthly temperature series, and the compiled corresponding daily series 1898–2018, Polar Res., 39, 3614, http://dx.doi.org/10.33265/polar.v39.3614, 2020.

4. We would also like to see more inferences or insights in the discussion that are closely related to the results or conclusion of this paper. For example, we are curious if the results of 1767-68 being warmer than now and 1784-92 being colder than now are reliable, over what range, and whether there is similar corroboration in other regions. As well, is this phenomenon just indicating exceptional years, or is it actually somewhat indicating that the climate becoming colder during 20 years (which means the end of the warm period in 18th century). We believe that such issue to be discussed can enhance the significance of the whole study.

ANS: Thank you for this comment and suggestion. Yes, we agree that it is a little surprising that the year 1767/68 belonging to the Little Ice Age period was significantly warmer than the present reference period. But this is only one year, and it is not really exceptional that so high temperatures occurred. The lack of data does not allow us to check whether this year was as warm in other areas of the Arctic, especially in Greenland. But we have at our disposal short synthetic descriptions of weather for each month written by Cranz (1820). Analysis of these descriptions indicates the occurrence of exceptionally mild weather in the study historical year in many months. For example, he wrote that mild weather conditions occurred in the cold half-year in December, March and particularly in January. About the latter month Cranz wrote: "*This month, which in Germany was colder than in the year 1740, was in Greenland remarkably mild.*" Also, it was warm from April to July, but especially in June. The latter month was summarised by him as follows: "*The air of this month was generally mild, excepting some cloudy forenoon: there was almost constant sunshine and agreeable spring weather, which is rare in Greenland.*" If the thermometer was not correctly protected against the sun rays (at this time screens were not used) such weather (i.e. very sunny weather) could have caused some warm bias in measurements. Warm bias in June was also probably partly connected with the relocation of the measurement site to Pissiksarbik, where, according to Cranz (1820) "*... the sun's rays are more powerful*". In addition, we also checked whether in present times such warm years as the historical year 1767/68 have occurred. Calculations of mean temperature in Nuuk for a comparable period (Sep–May) revealed that in the contemporary period (Sep 1990–May 2020) warmer years than in 1767/68 (-1.92 °C) have occurred three times: 2009/10 (-0.96 °C), 2003/04 (-1.78 °C) and 2012/13 (-1.90 °C).

We sincerely hope that our suggestions can better improve the quality of manuscripts, which we believe to be of great scholarly value.

ANS: Thank you for all your valuable suggestions. We hope that we have answered them satisfactorily. We confirm that all suggestions were very helpful in improving the readability of the paper.

---

## Author Comment (AC2)

Journal: Climate of the Past

Manuscript ID: cp-2023-95

Title: Air temperature changes in SW Greenland in the second half of the 18th century

Dear Chandal Camendish

Editor, Climate of the Past

We would like to thank the 2nd anonymous reviewer for providing positive feedback and constructive comments on our manuscript. All comments were carefully considered, and we believe they helped us improve the description of our work. The detailed corrections/modifications are listed below, point by point.

(Note: The changes in the text and the answers to the reviewer's questions/suggestions are marked in red font. We revised the text, taking into account all comments and suggestions proposed by the reviewer. All changes have been carefully applied to the text.)

**Referee No. 2.**

This is a rigorous and deep work, well organized in general, done by authors with ample experience in the study of Arctic climatology, who, by delving into historical data, try to obtain the maximum information available. They are aware of the testimonial value of the original data and of the possibility that they could be used as a reference for subsequent local and regional climate reconstruction studies.

ANS: Thank you.

This study presents the results of an analysis of historical climate data referring to the second half of the 18th century in Greenland. It covers two groups of years: 1767-68 and 1784-1792. The analysis is focused especially on the second of these groups, because it contains the most data. Although there is no complete information for all the years and months included in the latter group, the results obtained are in line with those of other studies in Greenland and the Arctic which demonstrates both the quality of the data used and the reliability of the analysis. The temporal sequence analyzed is also a significant contribution, because no data as old as these exist for the Arctic area to date and, as in all historical data prior to contemporaneous observations, there is no option but to study what is available.

ANS: Thank you.

The general guidelines I followed for the review were the following: Control of the general structure of the work and the contents of the different sections. Proposals to maximize the value of the analysis carried out and to enhance the results. Proposals to expand interest in the work in non-specialist readers along with reaching other related scientific fields

The statistical treatment performed is based on the comparison of historical data with current data from the 1991-2020 series by referencing historical data to the 1991-2020 series and checking

deviations from normal value. This is a type of analysis that fits well with the nature of the data and is accompanied by numerous graphs and tables.

ANS: Thank you.

SPECIFIC COMMENTS

Area and method section

This section is too short considering the interest and the complexity of the study's reconstructions, methods used, or motivations to start this analysis. To complete it and help readers to assimilate the text rationally and fluently, the following is proposed:

The text would improve by including some comment on the study area, considering its geographical singularity in relation to the meteorology, climatology, and natural environment.

ANS: Thank you very much for this suggestion. We have added significantly more information about the natural environment and climatology of the study area in the text.

For both historical and statistical interest, it would be quite convenient to include more information about the methodology used to obtain the original data as well as its preparation and correction.

ANS: Thank you very much for this suggestion. We have added more information about the methodology which we used in preparing the data for the analysis, for details see the revised version of the manuscript.

Even though the statistical methods selected fit well with the typology of the original data, please add some comments to justify, in climatic or/and environmental terms, why the indexes in table 2 were selected.

ANS: We answered exactly the same suggestion directed to us by the first reviewer. The answer is given in Reply to Reviewer 1, point 3.

Discussion section

The comparison of the results with the previous references is intense and complete. However, it is focused on contrasting it with other authors' results, relegating other interesting findings. Considering that the study is based on historical data not contained in any previous work, and that unusual statistical indices have been used, this section should also include an interpretation of the complete statistical work that has been done, as well as its climatic, environmental, and human implications. Consequently, these should be reflected in the conclusions section.

Conclusion section

A part of the paragraph between lines 439 and 450 in page 21 could be moved to section 2.

ANS: Thank you very much for this suggestion. After discussion among authors we came to the opinion that, although your proposition is good, we think that it will be a little better if we leave this paragraph here as it is, and will instead add some information about the possible biases in section 2, as you propose. One reason for this decision is that many scientists don't initially read the entire paper but concentrate on the abstract and conclusion.

TECHICAL CORRECTIONS

Section numbering

The Discussion section numbering should be 4 and, consequently, the Conclusion section 5.

ANS: corrected

Repeated paragraphs

Page 12, paragraph 2, lines 245-255 is repeated, as well as pages 14-15, lines 274-282.

ANS: Thank you very much for this suggestion. We have already also noted this repetition.

The second repeated passage was deleted.

References missed in the Reference list.

Bertrand et al. (2002)

Born et al. (2021)

Kaufman et al. (2009)

Houghton et al (1990)

Overpeck et al. (1997)

ANS: Thank you very much for this information. All missing references were added.

3. TYPING ERRORS

Lines 339-340: delete space between; and Kobashi.

ANS: corrected

Line 340: Publication year in Crespin et al. no coincidence between text and reference list. In line 349: Crespin et al. (2012), in line 378 Crespin et al. (2009) and Crespin let al. (2014), in line 380Crespin et al. (2009), but in the reference list: Crespin et al. (2014) (2013) and (2019).

ANS: corrected

Lines 395 correct year 1021 by 1921

ANS: 1021 is the corrected date, because the value of the temperature anomaly was calculated for the Medieval Warm Period, 1021-1050.

Line 460 The Author contribution section name is repeated and wrong in this place.

ANS: corrected

Line 471: Delete , after Corne

ANS: corrected

Line 478.  Does [et]. c. means [etc.]?

ANS: corrected

Line 486: delete point after Ebers

ANS: corrected

Line 550: replace the year 2012 by 2021

ANS: corrected

Line 551: delete , after . in Lüdecke, C.

ANS: corrected

Line 606: add publication year

ANS: corrected